# Four-dimensional mesospheric and lower thermospheric wind fields using Gaussian process regression on multistatic specular meteor radar observations

Ryan Volz[1], Jorge L. Chau[2], Philip J. Erickson[1], Juha P. Vierinen[3], J. Miguel Urco[2], and Matthias Clahsen[2]

[1]Haystack Observatory, Massachusetts Institute of Technology, USA
[2]Leibniz Institute of Atmospheric Physics at the University of Rostock, Germany
[3]UiT Arctic University of Norway, Norway

**Correspondence:** Ryan Volz (rvolz@mit.edu)

**Abstract.** Mesoscale dynamics in the mesosphere and lower thermosphere (MLT) region have been difficult to study from either ground- or satellite-based observations. For understanding of atmospheric coupling processes, important spatial scales at these altitudes range between tens to hundreds of kilometers in the horizontal plane. To date, this scale size is challenging observationally, and so structures are usually parameterized in global circulation models. The advent of multistatic specular meteor radar networks allows exploration of MLT mesoscale dynamics on these scales using an increased number of detections and a diversity of viewing angles inherent to multistatic networks. In this work, we introduce a four dimensional wind field inversion method that makes use of Gaussian process regression (GPR), a non-parametric and Bayesian approach. The method takes measured projected wind velocities and prior distributions of the wind velocity as a function of space and time, specified by the user or estimated from the data, and produces posterior distributions for the wind velocity. Computation of the predictive posterior distribution is performed on sampled points of interest and is not necessarily regularly sampled. The main benefits of the GPR method include this non-gridded sampling, the built-in statistical uncertainty estimates, and the ability to horizontally-resolve winds on relatively small scales. The performance of the GPR implementation has been evaluated on Monte Carlo simulations with known distributions using the same spatial and temporal sampling as one day of real meteor measurements. Based on the simulation results we find that the GPR implementation is robust, providing wind fields that are statistically unbiased and with statistical variances that depend on the geometry and are proportional to the prior velocity variances. A conservative and fast approach can be straightforwardly implemented by employing overestimated prior variances and distances, while a more robust but computationally intensive approach can be implemented by employing training and fitting of model hyperparameters. The latter GPR approach has been applied to a 24-hour data set and shown to compare well to previously used homogeneous and gradient methods. Small scale features have reasonably low statistical uncertainties, implying geophysical wind field horizontal structures as low as 20-50 km. We suggest that this GPR approach forms a suitable method for MLT regional and weather studies.

## 1 Introduction

The mesoscale neutral dynamics of the mesosphere and lower thermosphere (MLT) region are challenging to study, despite their importance in global circulation models. Due to the lack of observations, these scales are usually parameterized in models (e.g., Liu, 2019). MLT large scale dynamics have been studied with monostatic specular meteor radars (SMRs) by providing mean horizontal winds over areas with approximately 200-300 km radius at MLT altitudes, and 1-2 hour and 2-3 km temporal and vertical resolutions, respectively (e.g. Hocking et al., 2001; Holdsworth et al., 2004). These measurements have made significant contributions to community understanding of the climatological behavior of mean winds, planetary waves, and total tides over a variety of SMR monostatic sites (e.g. Mitchell et al., 1999, 2002; Pancheva et al., 2002; Sandford et al., 2006; Hoffmann et al., 2010). Moreover, when the winds from more than one SMR widely separated in longitude at a similar latitude are combined, spatiotemporal ambiguities of tides and planetary waves have been successfully resolved (e.g., Murphy, 2003; Murphy et al., 2006; He et al., 2018; He and Chau, 2019). Monostatic SMRs have been also used to study MLT gravity wave momentum flux with wide and narrow beam observing configurations, with the caveat that spatial and temporal contributions are combined (e.g. Hocking, 2005; Fritts et al., 2012; Andrioli et al., 2013; Placke et al., 2015).

Recently, multistatic configurations have been proposed to complement these previous studies and to allow the investigation of MLT mesoscale dynamics. These configurations include the MMARIA (Multi-static Multi-frequency Agile Radar Investigations of the Atmosphere) concept (Stober and Chau, 2015; Chau et al., 2017). This concept has been further augmented by the SIMONe (Spread Spectrum Interferometric Multistatic meteor radar Observing Network) approach (Chau et al., 2019). By using recent technological developments in atmospheric radars, such as spread-spectrum, MIMO (Multi-input multiple-output), and compressed sensing approaches (Vierinen et al., 2016; Urco et al., 2018, 2019b), SIMONe allows the implementation of MMARIA with several attractive qualities: it is easier, cheaper, and inherently expandable compared to original proposed configurations using traditional pulsed systems. Examples of SIMONe implementations in Germany, Peru and Argentina can be found in several studies (Vierinen et al., 2019; Charuvil Asokan et al., 2020; Vargas et al., 2020; Chau et al., 2021; Conte et al., 2021).

Multistatic observing approaches allow a large increase in scattering detections per unit time along with observation of the same volume from different viewing angles. These two features unlock the possibility of estimating the spatial features of the wind within the observed volume. Depending on the resolutions and spatial scales covered, different aspects of MLT mesoscale dynamics and coupling studies can be studied with the technique. For example, at scales between a few tens of kilometers to a few hundreds of kilometers, the contributions of gravity waves and strongly stratified turbulence can be studied with multistatic approaches (e.g. Roberts and Larsen, 2014; Marino et al., 2015).

The spatial structure of horizontal winds has been also pursued using a variety of other techniques including meteorological radars in the lower atmosphere, coherent scatter radars in the mesosphere, and Fabry-Perot interferometers in the thermosphere (e.g., Browning and Wexler, 1968; Chau et al., 2017; Meriwether et al., 2008). As in the case of the initial MMARIA analysis, these techniques typically approximate wind fields as analytic, differentiable polynomials in order to obtain gradients of the horizontal winds. Although they provide additional spatial information beyond direct single-point information, these methods

can aggressively smooth real spatial structure and, in some cases, can introduce artificial structure, particularly in regions with sparse or noisy measurements. In recent years, a variety of analysis approaches using statistical inverse theory have been applied to these and similar problems. These studies have the goal of estimating the spatial structure of multi-point projected wind velocities and electric fields (e.g., Nicolls et al., 2014; Hysell et al., 2014; Harding et al., 2015; Stober et al., 2018). For example, a Tikhonov regularization originally developed for a optical network of Fabry-Perot Interferometers (Harding et al., 2015) has been adapted to yield MLT wind fields over Peru (Chau et al., 2021).

As in any statistical inverse theory problem, more independent samples are desirable to reduce the impact of regularization constraints and to improve the quality of the estimates. In November 2018, a short observing campaign was conducted in northern Germany, herein denoted SIMONe2018, in which six existing MMARIA links were complemented with eight additional SIMONe links. During this campaign, we obtained on average two hundred thousand meteor scatter observations per day (e.g., Vierinen et al., 2019; Charuvil Asokan et al., 2020). For reference, a monostatic SMR obtains on average ten thousand meteors per day at a comparable latitude and seasonal time.

Some previous analysis methods have been published on multistatic observations of MLT mesoscale dynamics, such as the gradient method and variants of Tikhonov regularization (Chau et al., 2017; Stober et al., 2018; Chau et al., 2021). However, given the novelty of multistatic measurements and the lack of a reliable ground truth observation, different wind field approaches still need to be explored, particularly in the properties of resulting statistical measure bias and variance. In this work, we introduce a multistatic analysis technique based on Gaussian process regression (GPR) (Rasmussen and Williams, 2006). Some of the main benefits of GPR are that analysis predictions essentially interpolate the measurements (within error bounds) and that final output products inherently include quantitative uncertainties.

GPR is a Bayesian and non-parametric approach currently being used in many different machine learning applications (e.g., Wahlström et al., 2013; Foreman-Mackey et al., 2017). As a Bayesian technique, a key user input is the specification of a prior distribution for the values to be estimated, including hyperparameters of the distribution. Despite needing these hyperparameters, GPR is non-parametric in the sense that it does not compress the training data into a finite-dimensional parameter vector, in contrast to parametric methods like linear regression (Rasmussen and Williams, 2006). GPR is also known in other fields as *kriging*, and it has a long history of use in geostatistics under that name (Matheron, 1973; Journel and Huijbregts, 1978; Daley, 1991). Deep connections can be found between GPR and interpolation techniques using reproducing kernel Hilbert spaces (Scheuerer et al., 2013), including those that employ regularization. This ties GPR mathematically to the previously-mentioned wind field estimation techniques, but the Bayesian viewpoint afforded by GPR can be more natural for expressing prior information and analyzing uncertainty. We direct the reader to Rasmussen and Williams (2006) for a general discussion of GPR and its place in the wider estimation landscape.

In this article, we start by introducing the wind estimation problem and geometrical considerations. Next, we present the wind field estimation method using GPR, including the necessary mathematical expressions. The proposed estimation is subsequently applied to both Monte Carlo simulations and to measurements from the SIMONe2018 campaign in Sect. 5 and 6, respectively. In the latter section, estimated wind fields are compared to the winds obtained with the homogeneous and gradient

methods, i.e., to the zero- and first-order Taylor expansion approximations. Finally, we discuss the benefits and challenges of the proposed estimation approach for MLT wind field studies.

## 2 Specular meteor radar measurements and geometry

SMRs receive echoes from meteor trails when the radar Bragg vector ($\boldsymbol{k}_B$) points perpendicular to them. The Doppler shift ($f$) of the received signal of a meteor echo at time $t$ and location given by longitude, latitude, and altitude ($\Lambda, \Phi, z$) results from the projection of the atmospheric wind vector ($\boldsymbol{u}$) in the Bragg vector $\boldsymbol{k}_B$ (e.g, Hocking et al., 2001; Holdsworth et al., 2004), i.e.,

$$f(\Lambda, \Phi, z, t) = \frac{1}{2\pi} \begin{bmatrix} k_u & k_v & k_z \end{bmatrix} \begin{bmatrix} u(\Lambda, \Phi, z, t) \\ v(\Lambda, \Phi, z, t) \\ w(\Lambda, \Phi, z, t) \end{bmatrix} \tag{1}$$

where $k_u$, $k_v$, and $k_w$ are the Bragg vector components of $\boldsymbol{k}_B$ and $u$, $v$, and $w$ are wind vector components of $\boldsymbol{u}$ in the zonal (East), meridional (North), and vertical (Up) directions, respectively. The Bragg vector is given by the difference of the scattered and incident wavevectors, i.e., $\boldsymbol{k}_B = \boldsymbol{k}_s - \boldsymbol{k}_i$. Using interferometry on reception, the angle of arrival (AOA) is obtained. In the case of MIMO systems, interferometry is also implemented on transmission, allowing measurement of the angle of departure (AOD) (e.g., Chau et al., 2019). By combining these angles along with range information, the meteor location ($\Lambda, \Phi, z$) and Bragg wavevectors are obtained. In the reductive case of monostatic systems, $\boldsymbol{k}_B = -2\boldsymbol{k}_i$ and its magnitude is equal to $4\pi/\lambda$, where $\lambda$ is the radar wavelength.

As mentioned above, traditionally a mean horizontal wind has been obtained from analysis that simultaneously solves $N$ equations of the form of (1), with the assumption that the wind is constant in the observed volume (zero-order Taylor approximation or homogeneous method). The data for the $N$-equation set was obtained by binning desired observations with regular altitude and temporal resolutions. In general, with a sufficient number of meteors and viewing angles, the method yields spatial information of the wind inside the observed volume. For example, Chau et al. (2017) has implemented a gradient method, where the wind field estimation includes the first-order Taylor expansion terms.

In multistatic geometries, both the observed volumes and separations of the multi-static links are relatively large. For this reason, it is necessary to take the Earth's geoid shape into account. Moreover, the GPR model described in the next section is directly dependent on calculating coordinate distances accurately. This implies that altitudes and horizontal distances that account for the Earth's curvature, the measurement goal, must also try to minimizing mapping distortions, particularly in distance scaling. Use of a naive geometric projection such as the equirectangular projection, in which latitude and longitude are simply scaled to yield $x-y$ coordinates in meters, does not satisfy these requirements. Therefore, in this work, we use a local azimuthal equidistant projection centered in the observing region, with Earth shape based on the well known WGS84 geoid model. This projection is used to transform longitude and latitude into local $x$ and $y$ coordinates, where horizontal distance in $x$ and $y$ reasonably approximates the true geodesic distance. Subsequently, we use these $(x, y)$ projected coordinates in place of

$(\Lambda, \Phi)$ geodetic coordinates from (1). Note that this does not change the definitions of $(u, v, w)$ and $\boldsymbol{k}_B$, which remain aligned with a local East-North-Up coordinate system and not, in general, with the projected $x$ and $y$ coordinates.

To represent a set of Doppler wind measurements, we use the following notation for the measurement equation. Let $\boldsymbol{x}_m = (t_m, z_m, y_m, x_m)$ denote the coordinates for a measurement $m$ of $M$. Then the ensemble of coordinates is given by the matrix $\mathbf{X}$ as

$$
\quad \mathbf{X} = \begin{bmatrix} \boldsymbol{x}_1^\mathsf{T} \\ \vdots \\ \boldsymbol{x}_M^\mathsf{T} \end{bmatrix} = \begin{bmatrix} t_1 & z_1 & y_1 & x_1 \\ \vdots & \vdots & \vdots & \vdots \\ t_M & z_M & y_M & x_M \end{bmatrix}, \tag{2}
$$

and the corresponding wind vectors are given by

$$
\boldsymbol{u} = \begin{bmatrix} u(\boldsymbol{x}_1) \\ \vdots \\ u(\boldsymbol{x}_M) \end{bmatrix} \qquad \boldsymbol{v} = \begin{bmatrix} v(\boldsymbol{x}_1) \\ \vdots \\ v(\boldsymbol{x}_M) \end{bmatrix} \qquad \boldsymbol{w} = \begin{bmatrix} w(\boldsymbol{x}_1) \\ \vdots \\ w(\boldsymbol{x}_M) \end{bmatrix}. \tag{3}
$$

We group the Bragg vectors of a set of measurements by component and combine with the $\frac{1}{2\pi}$ scaling to give $u$, $v$, and $w$ measurement vectors:

$$
\quad \boldsymbol{a}_u = \frac{1}{2\pi} \begin{bmatrix} k_{u_1} \\ \vdots \\ k_{u_M} \end{bmatrix} \qquad \boldsymbol{a}_v = \frac{1}{2\pi} \begin{bmatrix} k_{v_1} \\ \vdots \\ k_{v_M} \end{bmatrix} \qquad \boldsymbol{a}_w = \frac{1}{2\pi} \begin{bmatrix} k_{w_1} \\ \vdots \\ k_{w_M} \end{bmatrix} \tag{4}
$$

Finally, using $\odot$ to denote the element-wise (Hadamard) vector product, our measurement equation following from (1) for the ensemble of Doppler measurements $\boldsymbol{f}$ is

$$
\boldsymbol{f}(\mathbf{X}) = \begin{bmatrix} f(\boldsymbol{x}_1) \\ \vdots \\ f(\boldsymbol{x}_M) \end{bmatrix} = \boldsymbol{a}_u \odot \boldsymbol{u} + \boldsymbol{a}_v \odot \boldsymbol{v} + \boldsymbol{a}_w \odot \boldsymbol{w} + \boldsymbol{\epsilon} \tag{5}
$$

where $\boldsymbol{\epsilon} \sim \mathcal{N}(0, \boldsymbol{\Sigma}_n)$ is zero-mean Gaussian measurement uncertainty with covariance $\boldsymbol{\Sigma}_n$.

## 135   3   Estimation Problem

The estimation task is to take a set of Doppler measurements $\boldsymbol{f}$ and infer wind values $u(\boldsymbol{x}')$, $v(\boldsymbol{x}')$, $w(\boldsymbol{x}')$ at a chosen location $\boldsymbol{x}'$ using the measurement model from (5). We employ Gaussian process regression (GPR) to model the winds and hence Doppler measurements as a stochastic process. This approach allows estimation at arbitrary coordinates (convenient for the random meteor locations and non-gridded prediction) and produces statistical uncertainty as an output product.

Our GPR method is implemented as a 3-staged process. First, one defines the form for the model, which includes mean and covariance functions and their hyperparameters. Then, one fully specifies the model by setting hyperparameter values, either

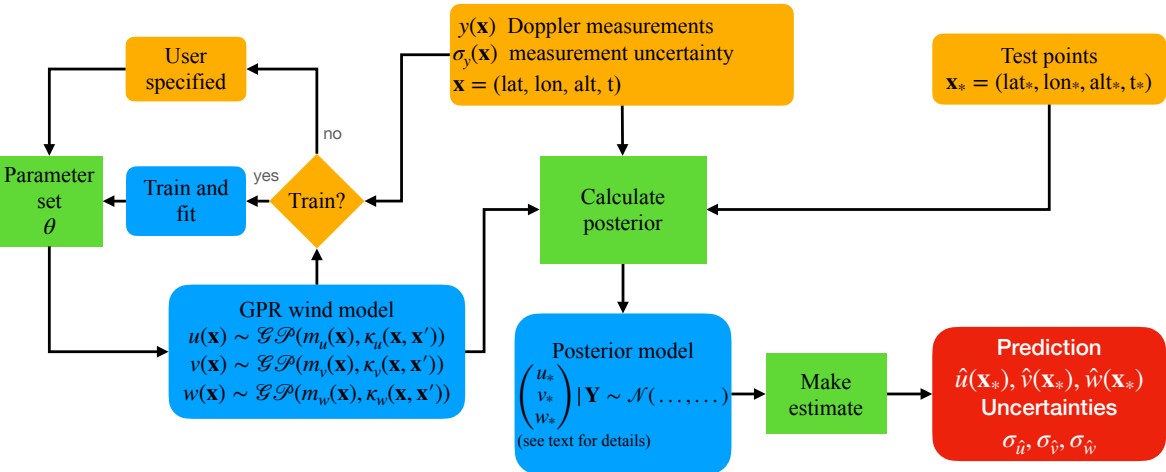

**Figure 1.** *Block diagram of processing flow.* The blocks in orange indicate input from the user, blocks in green belong to the GPR model, and the estimates are obtained in the red block (see text for details).

through prior knowledge or a separate fitting process. Finally, one applies the specified model to a set of measurements to calculate the posterior predictive distribution and make an estimate at points of interest. Figure 1 summarizes our implementation in a block diagram. In the following paragraphs, we describe the method in detail.

### 3.1 Gaussian process definitions

For a function $f(\boldsymbol{x})$ drawn from a Gaussian process, we write

$$f(\boldsymbol{x}) \sim \mathcal{GP}\big(m(\boldsymbol{x}), \kappa(\boldsymbol{x}, \boldsymbol{x'})\big). \tag{6}$$

This representation is fully defined by mean and covariance functions which describe the first- and second-order statistics:

$$m(\boldsymbol{x}) = \mathbf{E}\big[f(\boldsymbol{x})\big] \tag{7}$$

$$\kappa(\boldsymbol{x}, \boldsymbol{x'}) = \mathbf{E}\Big[\big(f(\boldsymbol{x}) - m(\boldsymbol{x})\big)\big(f(\boldsymbol{x'}) - m(\boldsymbol{x'})\big)\Big] \tag{8}$$

where $\mathbf{E}$ denotes expected value. Gaussian processes are convenient because evaluating them at a set of points leads to a Gaussian random vector

$$
\begin{bmatrix} f(\boldsymbol{x}_1) \\ \vdots \\ f(\boldsymbol{x}_N) \end{bmatrix} \sim \mathcal{N}\left( \begin{bmatrix} m(\boldsymbol{x}_1) \\ \vdots \\ m(\boldsymbol{x}_N) \end{bmatrix}, \begin{bmatrix} \kappa(\boldsymbol{x}_1,\boldsymbol{x}_1) & \cdots & \kappa(\boldsymbol{x}_1,\boldsymbol{x}_N) \\ \vdots & \ddots & \vdots \\ \kappa(\boldsymbol{x}_N,\boldsymbol{x}_1) & \cdots & \kappa(\boldsymbol{x}_N,\boldsymbol{x}_N) \end{bmatrix} \right) \tag{9}
$$

which enables tractable computation. We recast this compactly using matrix notation as

$$
\boldsymbol{f}(\mathbf{X}) \sim \mathcal{N}(\boldsymbol{m}(\mathbf{X}), \mathbf{K}(\mathbf{X},\mathbf{X})). \tag{10}
$$

It might seem like this model is too simple to be useful, but Gaussian processes have a lot of flexibility to fit a wide variety of functions because the posterior distribution is constructed non-parametrically and directly incorporates the measurements. Additionally, a modeler has a lot of freedom in applying Gaussian processes by choosing the form of the mean and covariance functions, including specifying hyperparameters.

### 3.2 Wind component prior distributions

Since we want to estimate the wind components, we model them as independent Gaussian processes:

$$
u(\boldsymbol{x}) \sim \mathcal{GP}(m_u(\boldsymbol{x}), \kappa_u(\boldsymbol{x}, \boldsymbol{x}')) \tag{11}
$$

$$
v(\boldsymbol{x}) \sim \mathcal{GP}(m_v(\boldsymbol{x}), \kappa_v(\boldsymbol{x}, \boldsymbol{x}')) \tag{12}
$$

$$
w(\boldsymbol{x}) \sim \mathcal{GP}(m_w(\boldsymbol{x}), \kappa_w(\boldsymbol{x}, \boldsymbol{x}')). \tag{13}
$$

Assuming Gaussianity of the wind processes is not simply for convenience (although it does enable closed-form computation). Given some mean and covariance, a Gaussian distribution has the maximum entropy (Cover and Thomas, 2006). In other words, assuming normality imposes the minimal prior information about the wind processes within a second-order statistical framework. The winds likely have more structure than this, including cross-covariances between the components, but this assumption ensures conservative estimates without prior knowledge of the true statistical structure of the wind processes.

Many choices for the mean functions are possible, but for simplicity we restrict our attention to means that are fixed without tunable hyperparameters. Even under this restriction, one can use a standard parametric model for the mean functions, and as long as the parameter-fitting is done with linear regression prior to GPR analysis, no additional hyperparameters are added to the GPR model. In general, the mean functions have less impact on the GPR results than the covariance functions, and we will see later how the posterior predictive distribution is more strongly driven by the measurements and the covariance functions. Often a zero mean is sufficient to produce good results (Rasmussen and Williams, 2006), and that holds for this case as well. Nevertheless, the mean can be useful for including well-known effects. In the models for subsequent sections, we have used two cubic splines taken as a tensor product over altitude and time to produce a mean that accounts for large-scale tidal components.

For the covariance functions, we choose a functional form where each wind component has an independent amplitude multiplying a common distance kernel:

$$\kappa_u(\boldsymbol{x}, \boldsymbol{x}') = \sigma_u^2 \kappa_d(\boldsymbol{x}, \boldsymbol{x}') \tag{14}$$

$$\kappa_v(\boldsymbol{x}, \boldsymbol{x}') = \sigma_v^2 \kappa_d(\boldsymbol{x}, \boldsymbol{x}') \tag{15}$$

$$\kappa_w(\boldsymbol{x}, \boldsymbol{x}') = \sigma_w^2 \kappa_d(\boldsymbol{x}, \boldsymbol{x}'). \tag{16}$$

Using a common distance kernel is convenient for simplifying computations, and we expect that relaxing this assumption in the future would allow for increased expressiveness at the cost of computational burden. The distance kernel $\kappa_d$ is chosen to be the Matérn covariance with $\nu = \frac{5}{2}$, using length scales given by $\delta_t$, $\delta_z$, $\delta_y$, and $\delta_x$ for the coordinate dimensions:

$$\kappa_d(\boldsymbol{x}, \boldsymbol{x}') = \left(1 + \sqrt{5}r + \frac{5}{3}r^2\right)e^{-\sqrt{5}r} \tag{17}$$

with

$$r = \left\|\frac{\boldsymbol{x} - \boldsymbol{x}'}{\boldsymbol{\delta}}\right\|_2 \tag{18}$$

$$\boldsymbol{\delta} = \begin{bmatrix} \delta_t & \delta_z & \delta_y & \delta_x \end{bmatrix}^{\mathsf{T}} \tag{19}$$

where $\|.\|_2$ represents the Euclidean norm. Altogether, this results in a hyperparameter set $\boldsymbol{\theta}$ of

$$\boldsymbol{\theta} = \begin{bmatrix} \sigma_u^2 & \sigma_v^2 & \sigma_w^2 & \delta_t & \delta_z & \delta_y & \delta_x \end{bmatrix}^{\mathsf{T}} \tag{20}$$

for the GPR wind model. We chose the Matérn-$\frac{5}{2}$ covariance because it is twice-differentiable but not infinitely-differentiable, so it provides relatively smooth functions while still allowing for rapid, geophysically-driven changes that might be expected in wind fields. It is a typical choice for physical processes for this reason across a wide series of applications (Rasmussen and Williams, 2006).

Jointly and in matrix notation, we then write the Gaussian random vectors for the winds at a set of points $\mathbf{X}$ as

$$\begin{bmatrix} \boldsymbol{u} \\ \boldsymbol{v} \\ \boldsymbol{w} \end{bmatrix} \sim \mathcal{N}\left(\begin{bmatrix} \boldsymbol{m}_u(\mathbf{X}) \\ \boldsymbol{m}_v(\mathbf{X}) \\ \boldsymbol{m}_w(\mathbf{X}) \end{bmatrix}, \begin{bmatrix} \mathbf{K}_u(\mathbf{X}, \mathbf{X}) & \mathbf{0} & \mathbf{0} \\ \mathbf{0} & \mathbf{K}_v(\mathbf{X}, \mathbf{X}) & \mathbf{0} \\ \mathbf{0} & \mathbf{0} & \mathbf{K}_w(\mathbf{X}, \mathbf{X}) \end{bmatrix}\right). \tag{21}$$

Note that since we have defined the wind component processes independently, the cross terms are zero in the joint covariance matrix. However, this is not to say that we strictly enforce zero cross-covariance between the wind terms with this model. Rather, it is more accurate to say that we do not require prior knowledge of the cross-covariance but also cannot benefit from the improved estimation that such knowledge would provide.

### 3.3 Doppler measurement prior distribution

Since we are taking the wind components as Gaussian processes, and (5) provides a linear relationship between the wind components and Doppler measurements, the Doppler measurements themselves also take the form of a Gaussian process. For

a set of measurements $\boldsymbol{f}$ corresponding to the locations $\mathbf{X}$, this produces a formulation as

$$\boldsymbol{f} \sim \mathcal{N}(\boldsymbol{m}_f(\mathbf{X}), \mathbf{K}_f(\mathbf{X}, \mathbf{X})) \tag{22}$$

where

$$\boldsymbol{m}_f(\mathbf{X}) = \boldsymbol{a}_u \odot \boldsymbol{m}_u(\mathbf{X}) + \boldsymbol{a}_v \odot \boldsymbol{m}_v(\mathbf{X}) + \boldsymbol{a}_w \odot \boldsymbol{m}_w(\mathbf{X})$$

$$\mathbf{K}_f(\mathbf{X}, \mathbf{X}) = (\boldsymbol{a}_u \boldsymbol{a}_u{}^{\mathsf{T}}) \odot \mathbf{K}_u(\mathbf{X}, \mathbf{X}) + (\boldsymbol{a}_v \boldsymbol{a}_v{}^{\mathsf{T}}) \odot \mathbf{K}_v(\mathbf{X}, \mathbf{X})$$

$$+ (\boldsymbol{a}_w \boldsymbol{a}_w{}^{\mathsf{T}}) \odot \mathbf{K}_w(\mathbf{X}, \mathbf{X}) + \boldsymbol{\Sigma}_n.$$

Note that the Gaussian process being measured is a linear composition. This is only a minor concern for our application, but it does make the formulation slightly different from the more typical examples. The following subsections provide the explicit formulas necessary to perform hyperparameter fitting and wind estimation using this model.

### 3.4 Model hyperparameter fitting

Fitting for the model hyperparameters $\boldsymbol{\theta}$ involves maximizing the likelihood function for the marginal distribution pertaining to a set of measurements. Assuming Doppler measurements $\boldsymbol{f}$ coming from the distribution defined in (22), the negative log-likelihood as a function of the hyperparameters is

$$-l(\boldsymbol{\theta}) = \frac{1}{2}(\boldsymbol{f} - \boldsymbol{m}_f)^{\mathsf{T}} \mathbf{K}_f{}^{-1}(\boldsymbol{f} - \boldsymbol{m}_f) + \frac{1}{2}\log \det \mathbf{K}_f - C \tag{23}$$

where $C$ is a fixed scaling constant. Minimizing this function requires evaluating the gradient of the negative log-likelihood. For each hyperparameter $\theta_i$, we thus have

$$\frac{\partial(-l(\boldsymbol{\theta}))}{\partial \theta_i} = \frac{1}{2}\operatorname{tr}\left((\boldsymbol{\alpha}\boldsymbol{\alpha}^{\mathsf{T}} - \mathbf{K}_f{}^{-1})\frac{\partial \mathbf{K}_f}{\partial \theta_i}\right) \tag{24}$$

where

$$\boldsymbol{\alpha} = \mathbf{K}_f{}^{-1}(\boldsymbol{f} - \boldsymbol{m}_f). \tag{25}$$

Continuing down the derivative chain for each type of hyperparameter produces

$$\frac{\partial \mathbf{K}_f}{\partial \sigma_i^2} = (\boldsymbol{a}_i \boldsymbol{a}_i{}^{\mathsf{T}}) \odot \mathbf{K}_d \tag{26}$$

$$\frac{\partial \mathbf{K}_f}{\partial \delta_i} = \left(\sigma_u^2(\boldsymbol{a}_u \boldsymbol{a}_u{}^{\mathsf{T}}) + \sigma_v^2(\boldsymbol{a}_v \boldsymbol{a}_v{}^{\mathsf{T}}) + \sigma_w^2(\boldsymbol{a}_w \boldsymbol{a}_w{}^{\mathsf{T}})\right) \odot \frac{\partial \mathbf{K}_d}{\partial \delta_i} \tag{27}$$

and

$$\frac{\partial \kappa_d(\boldsymbol{x}_j, \boldsymbol{x}_k)}{\partial \delta_i} = \frac{5}{3}(1 + \sqrt{5}r)e^{-\sqrt{5}r}\frac{1}{\delta_i}\left(\frac{(\boldsymbol{x}_j)_i - (\boldsymbol{x}_k)_i}{\delta_i}\right)^2 \tag{28}$$

where

$$r = \left\|\frac{\boldsymbol{x}_j - \boldsymbol{x}_k}{\boldsymbol{\delta}}\right\|_2. \tag{29}$$

With the objective and gradient known, fitting for $\boldsymbol{\theta}$ then involves feeding these functions into an appropriate optimization routine. We have observed the most reliable convergence using SciPy's implementation of the L-BFGS-B and SLSQP algorithms (Virtanen et al., 2020).

## 3.5 Wind estimation

Having defined the model hyperparameters either through fitting or prior specification, estimating the winds at a set of predic-
235 tion points $\mathbf{X}_*$ involves evaluating the posterior probability distribution given the measurements.

We start with the joint distribution between the measurements and the winds at the prediction points, which from previous definitions is given by:

$$\begin{bmatrix} \boldsymbol{f} \\ \boldsymbol{u}_* \\ \boldsymbol{v}_* \\ \boldsymbol{w}_* \end{bmatrix} \sim \mathcal{N} \left( \begin{bmatrix} \boldsymbol{m}_f(\mathbf{X}) \\ \boldsymbol{m}_u(\mathbf{X}_*) \\ \boldsymbol{m}_v(\mathbf{X}_*) \\ \boldsymbol{m}_w(\mathbf{X}_*) \end{bmatrix}, \mathbf{K}_{tot} \right) \tag{30}$$

where

$$
\quad \mathbf{K}_{tot} = \begin{bmatrix} \mathbf{K}_f(\mathbf{X},\mathbf{X}) & \boldsymbol{a}_u \odot \mathbf{K}_u(\mathbf{X},\mathbf{X}_*) & \boldsymbol{a}_v \odot \mathbf{K}_v(\mathbf{X},\mathbf{X}_*) & \boldsymbol{a}_w \odot \mathbf{K}_w(\mathbf{X},\mathbf{X}_*) \\ \mathbf{K}_u(\mathbf{X}_*,\mathbf{X}) \odot \boldsymbol{a}_u & \mathbf{K}_u(\mathbf{X}_*,\mathbf{X}_*) & \mathbf{0} & \mathbf{0} \\ \mathbf{K}_v(\mathbf{X}_*,\mathbf{X}) \odot \boldsymbol{a}_v & \mathbf{0} & \mathbf{K}_v(\mathbf{X}_*,\mathbf{X}_*) & \mathbf{0} \\ \mathbf{K}_w(\mathbf{X}_*,\mathbf{X}) \odot \boldsymbol{a}_w & \mathbf{0} & \mathbf{0} & \mathbf{K}_w(\mathbf{X}_*,\mathbf{X}_*) \end{bmatrix}. \tag{31}
$$

The posterior predictive distribution follows from conditioning on the measurements:

$$\begin{bmatrix} \boldsymbol{u}_* \\ \boldsymbol{v}_* \\ \boldsymbol{w}_* \end{bmatrix} \mid \boldsymbol{f} \sim \mathcal{N}(\boldsymbol{m}_{post}, \mathbf{K}_{post}) \tag{32}$$

where

$$
\boldsymbol{m}_{post} = \begin{bmatrix} \boldsymbol{m}_u(\mathbf{X}_*) \\ \boldsymbol{m}_v(\mathbf{X}_*) \\ \boldsymbol{m}_w(\mathbf{X}_*) \end{bmatrix} + \begin{bmatrix} \mathbf{K}_u(\mathbf{X}_*,\mathbf{X}) \odot \boldsymbol{a}_u \\ \mathbf{K}_v(\mathbf{X}_*,\mathbf{X}) \odot \boldsymbol{a}_v \\ \mathbf{K}_w(\mathbf{X}_*,\mathbf{X}) \odot \boldsymbol{a}_w \end{bmatrix} \mathbf{K}_f(\mathbf{X},\mathbf{X})^{-1}(\boldsymbol{f} - \boldsymbol{m}_f(\mathbf{X})) \tag{33}
$$

and

$$
\mathbf{K}_{post} = \begin{bmatrix} \mathbf{K}_u(\mathbf{X}_*,\mathbf{X}_*) & \mathbf{0} & \mathbf{0} \\ \mathbf{0} & \mathbf{K}_v(\mathbf{X}_*,\mathbf{X}_*) & \mathbf{0} \\ \mathbf{0} & \mathbf{0} & \mathbf{K}_w(\mathbf{X}_*,\mathbf{X}_*) \end{bmatrix}
$$

$$
- \begin{bmatrix} \mathbf{K}_u(\mathbf{X}_*,\mathbf{X}) \odot \boldsymbol{a}_u \\ \mathbf{K}_v(\mathbf{X}_*,\mathbf{X}) \odot \boldsymbol{a}_v \\ \mathbf{K}_w(\mathbf{X}_*,\mathbf{X}) \odot \boldsymbol{a}_w \end{bmatrix} \mathbf{K}_f(\mathbf{X},\mathbf{X})^{-1} \begin{bmatrix} \boldsymbol{a}_u \odot \mathbf{K}_u(\mathbf{X},\mathbf{X}_*) & \boldsymbol{a}_v \odot \mathbf{K}_v(\mathbf{X},\mathbf{X}_*) & \boldsymbol{a}_w \odot \mathbf{K}_w(\mathbf{X},\mathbf{X}_*) \end{bmatrix}. \tag{34}
$$

The mean of the posterior predictive distribution forms our estimate for the winds at the chosen points of interest, and this is given by

$$\hat{\boldsymbol{u}}(\mathbf{X}_*) = \mathbf{E}[\boldsymbol{u}_* \mid \boldsymbol{f}] = \boldsymbol{m}_u(\mathbf{X}_*) + (\mathbf{K}_u(\mathbf{X}_*, \mathbf{X}) \odot \boldsymbol{a}_u)\mathbf{K}_f(\mathbf{X}, \mathbf{X})^{-1}(\boldsymbol{f} - \boldsymbol{m}_f(\mathbf{X})) \tag{35}$$

$$\hat{\boldsymbol{v}}(\mathbf{X}_*) = \mathbf{E}[\boldsymbol{v}_* \mid \boldsymbol{f}] = \boldsymbol{m}_v(\mathbf{X}_*) + (\mathbf{K}_v(\mathbf{X}_*, \mathbf{X}) \odot \boldsymbol{a}_v)\mathbf{K}_f(\mathbf{X}, \mathbf{X})^{-1}(\boldsymbol{f} - \boldsymbol{m}_f(\mathbf{X})) \tag{36}$$

$$\hat{\boldsymbol{w}}(\mathbf{X}_*) = \mathbf{E}[\boldsymbol{w}_* \mid \boldsymbol{f}] = \boldsymbol{m}_w(\mathbf{X}_*) + (\mathbf{K}_w(\mathbf{X}_*, \mathbf{X}) \odot \boldsymbol{a}_w)\mathbf{K}_f(\mathbf{X}, \mathbf{X})^{-1}(\boldsymbol{f} - \boldsymbol{m}_f(\mathbf{X})). \tag{37}$$

Here we can see that the estimates near measurement locations, where $\mathbf{K}_{u,v,w}(\mathbf{X}_*, \mathbf{X})$ is large, are dominated by the prior covariance function specification. This is why the choice of prior covariance function is more important than the choice of prior mean function for making useful estimates and why our subsequent analysis is concentrated on the covariance hyperparameters.

Similarly, we obtain an estimate of the prediction uncertainty by using the posterior variance for each wind component, given by

$$\boldsymbol{\sigma}_{\hat{\boldsymbol{u}}}^2(\mathbf{X}_*) = \mathbf{Var}[\boldsymbol{u}_* \mid \boldsymbol{f}] = \sigma_u^2 - \mathrm{diag}\Big((\mathbf{K}_u(\mathbf{X}_*, \mathbf{X}) \odot \boldsymbol{a}_u)\mathbf{K}_f(\mathbf{X}, \mathbf{X})^{-1}(\boldsymbol{a}_u \odot \mathbf{K}_u(\mathbf{X}, \mathbf{X}_*))\Big) \tag{38}$$

$$\boldsymbol{\sigma}_{\hat{\boldsymbol{v}}}^2(\mathbf{X}_*) = \mathbf{Var}[\boldsymbol{v}_* \mid \boldsymbol{f}] = \sigma_v^2 - \mathrm{diag}\Big((\mathbf{K}_v(\mathbf{X}_*, \mathbf{X}) \odot \boldsymbol{a}_v)\mathbf{K}_f(\mathbf{X}, \mathbf{X})^{-1}(\boldsymbol{a}_v \odot \mathbf{K}_v(\mathbf{X}, \mathbf{X}_*))\Big) \tag{39}$$

$$\boldsymbol{\sigma}_{\hat{\boldsymbol{w}}}^2(\mathbf{X}_*) = \mathbf{Var}[\boldsymbol{w}_* \mid \boldsymbol{f}] = \sigma_w^2 - \mathrm{diag}\Big((\mathbf{K}_w(\mathbf{X}_*, \mathbf{X}) \odot \boldsymbol{a}_w)\mathbf{K}_f(\mathbf{X}, \mathbf{X})^{-1}(\boldsymbol{a}_w \odot \mathbf{K}_w(\mathbf{X}, \mathbf{X}_*))\Big). \tag{40}$$

Since the measurement covariance $\mathbf{K}_f$ term includes the assumed measurement noise, these equations effectively propagate the Doppler uncertainty through the measurement geometry and meteor density to produce the wind estimate uncertainty. However, we note that this uncertainty estimate ignores the cross terms in the covariance, both between test locations and among the wind components. These factors can also be included to give a more complete picture of how the individual estimates are correlated, at an increased computational cost. More detailed estimates could also be backed by a fully Bayesian approach that involves Markov chain Monte Carlo sampling of the posterior predictive distribution and includes full distributions for the hyperparameters $\boldsymbol{\theta}$.

Evaluating the posterior mean and covariance is a straightforward numerical linear algebra problem. However, given the potential sizes of the various covariance matrices, this can be computationally expensive. Mitigation of this implementation burden can be achieved with both matrix-free and approximate methods (e.g. Gardner et al., 2018; Wilson and Nickisch, 2015). Application of these methods are the subject of future work, but we note that their use would make practical fitting and evaluating more tractable.

## 4    SIMONe2018 Campaign

Before describing and presenting the simulation and experimental results, in this section we briefly describe the SIMONe2018 measurement campaign that was conducted in northern Germany between November 2nd and 9th, 2018. As mentioned in the Introduction, the SIMONe2018 campaign added eight SIMONe links to six existing MMARIA links. The MMARIA links consist of two pulsed transmitters located in Juliusruh (13.37°E, 54.63°N) and Collm (13.00°E, 51.31°N), operating at 32.55

and 36.2 MHz, respectively. The signals of these transmitters were received at four receiving stations located in Juliusruh, Neustrelitz (13.07°E, 53.33°N), Bornim (13.02°E, 52.44°N), and Collm, respectively.

For the SIMONe links, a coded continuous wave (CW) transmitter was operated from Kühlungsborn (11.77°E, 54.12°N) at 32.55 MHz. The transmitter array consisted of five two-element single polarization antennas, arranged in a Pentagon configuration. Each antenna transmitted a different pseudo-random code sequence, with 1000 bauds and 10 $\mu$s baud length. On reception, four single antennas were used, yielding MISO (multi-input single-output) links. In addition, the same 32.55 MHz antennas and receiving systems located in Neustrelitz and Bornim were used to receive the coded CW signals, forming both MISO and SIMO (single-input multiple-output) links at both sites.

The meteor signals from the pulsed links were detected and identified using a similar methodology as described in Hocking et al. (2001). In the case of the SIMONe links, the meteor signals were decoded and detected using the compressed sensing approach introduced by Urco et al. (2019a). Once the signals were detected, Doppler shift and interferometric angles were obtained from the autocorrelation and cross-correlation (between channels), respectively, in a similar manner as employed by Holdsworth et al. (2004). The interferometric angles were obtained using a combination of beam-forming and non-linear complex fitting of the time series data following Clahsen (2018) and Chau and Clahsen (2019), which includes estimating statistical uncertainties for the Doppler measurements. Such uncertainty estimates are used as quality checks or weights in fitting procedures. Location of the meteors and representation of the Bragg vector in the local meteor ENU coordinate system was performed using the WGS84 representation for an ellipsoidal Earth coordinate frame. That procedure has been described previously in Clahsen (2018) and Stober et al. (2018). More details of the SIMONe2018 campaign can be found in Vierinen et al. (2019) and Charuvil Asokan et al. (2020).

## 5 Monte Carlo Simulations

Monte Carlo simulations of the wind field ($\boldsymbol{u} = u, v, w$) are essential to gauge the bias and variance properties of the GPR method. To create realistic random wind fields with which we could simulate meteor measurements and compare the GPR estimate, we again made use of Gaussian processes. Instances of $\boldsymbol{u}(t, z, y, x)$ were drawn from the Gaussian random vector distribution described by (21) for specified sample locations, mean wind functions, and covariance amplitude and length scale hyperparameters. The hyperparameters used were as follows: $\sigma_u^2 = \sigma_v^2 = 900$ m$^2$ s$^{-2}$, $\sigma_w^2 = 90$ m$^2$ s$^{-2}$, $\delta_x = \delta_y = 50$ km, $\delta_z = 3$ km, and $\delta_t = 1800$ s. These velocity fields were used with observing geometries taken from one day of the SIMONe 2018 campaign, specifically November 5, 2018. At each real detection, the measured projected velocity was replaced by a new projected simulation velocity taking into account both the measured Bragg vector and the simulated $\boldsymbol{u}(t, z, y, x)$. In this way, we are able to test the proposed GPR method on actual measuring geometries.

Using the simulated measurements, we followed the GPR method from Sect. 3 to estimate the 4D wind field for comparison to the simulated winds. We explored fitting with different cubic spline forms for the mean wind functions, and qualitatively we found that the wind estimates were not sensitive to the details of the fit as long as it was reasonable. Even using a constant mean of zero produced qualitatively similar results. Thus, to remove a confounding variable, all of the estimation results

presented in this section use the exact mean functions that were used to simulate the winds, which in turn are the same mean functions fitted to the SIMON2018 data as described in Sect. 6. Likewise, we fit for the covariance hyperparameters from the simulated measurements and found that the results were similar (within 10%) to the values used for the simulation. This was reassuring and showed that the fitting procedure works at least when the winds can be described exactly by a Matérn-covariance Gaussian process. Similar to the mean, the estimated winds showed little qualitative sensitivity to small changes

in the covariance hyperparameters, so for the subsequent estimation results we used (as a baseline case) the same values for the amplitudes and length scales between the simulation and estimation Gaussian processes in order to remove fitting noise as a confounding variable. These comparisons should be viewed as a best-case scenario from the perspective of the model, and therefore they can be used primarily to explore the effects of meteor measurement spatial density and geometry on the quality of the wind estimates.

## 5.1   Qualitative comparison of horizontal winds

Figure 2 shows an example of results for simulated (left) and estimated (right) wind fields for three selected altitudes: 84, 90, and 96 km. The horizontal wind magnitude is color-coded (blue-green-yellow tones), while the direction is indicated by the over-plotted streamlines. The estimated values are also masked (altering transparency) in regions where the posterior predictive variances are high. Such regions are naturally where there are fewer meteor detections. Note that contrary to traditional methods

and despite the presentation here as horizontal slices, the estimates are not confined to a regular horizontal grid since solutions are inherently obtained in 4D. At an overall level, there is a very good agreement between the horizontal wind magnitude and direction at all altitudes in regions where the posterior predictive variance is reasonably low (full color areas).

## 5.2   Bias and error variance

For a more quantitative idea of the performance of the GPR method, we have repeated the Monte Carlo simulations 4700 times

using 100 instances at each $(t, z, y, x)$ location for 47 different overlapping time intervals throughout the day. This is equivalent to observing over 100 days with the same measurement statistics at each of the 47 time intervals of a given day. We estimated bias and error variance by calculating the sample mean and variance of the error between the estimated and simulated $u, v, w$ wind values over the $n = 4700$ time/trial instances. In the case of the horizontal winds, the bias is given as the magnitude of the mean error vector composed of both the zonal $u$ and meridional $v$ wind components, and the error variance is the sum of

both the $u$ and $v$ error variances.

    Figure 3 shows the bias of the horizontal wind error (left) color-coded with red tones and the error variance of the horizontal wind (right) color-coded with purple-yellow tones, in both cases for the same altitudes shown in Fig. 2. In the mean error panels, the posterior predictive variance is also indicated with green contours. A bias of less than $2\ \mathrm{m\,s^{-1}}$ is seen across the plots, and generally smaller biases are seen in the regions of lower predictive variance where there are more meteor detections. Note also

that the uncertainty contours (left) roughly match the shape of the actual error variance (right), which gives confidence that the uncertainty estimates are useful.

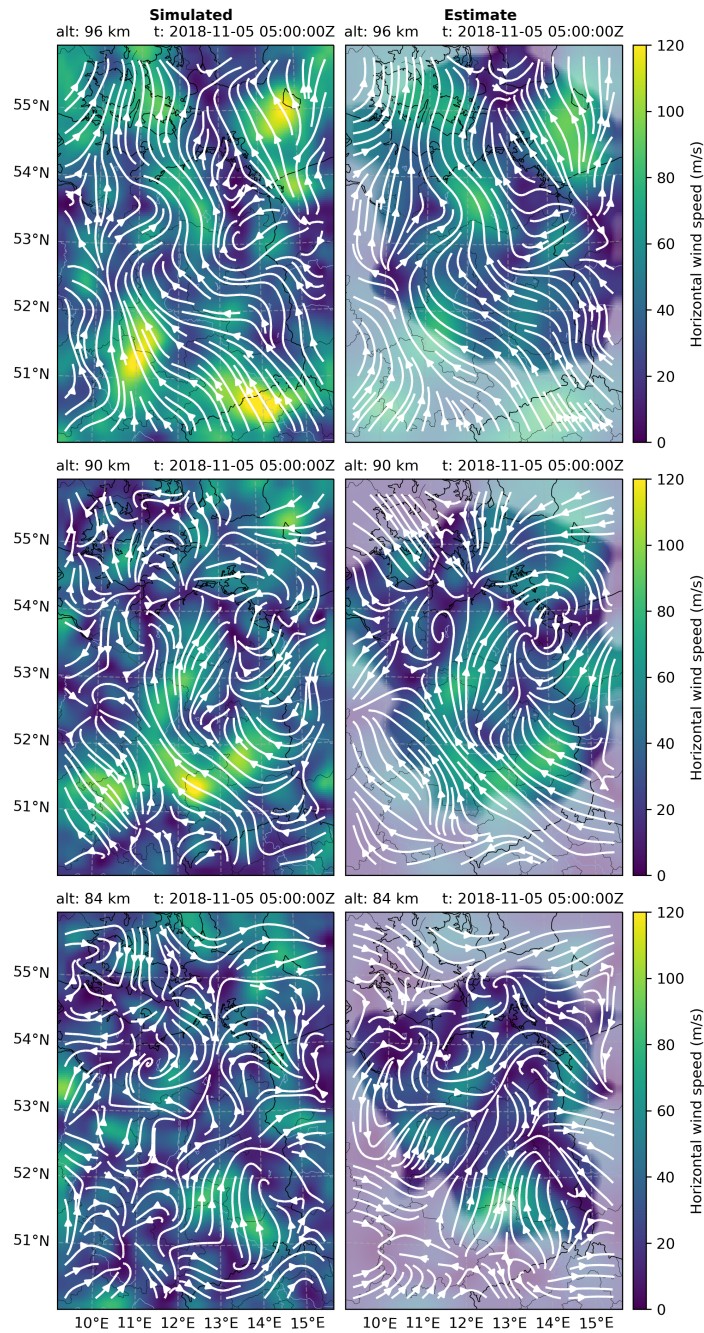

**Figure 2.** *Simulated wind field (left panels) compared to the resulting GPR estimate based on SIMONe-derived measurements (right panels).* Each panel shows the horizontal wind speed as a function of latitude and longitude overlaid by streamlines showing the wind flow. The estimated wind speed is masked at 50% transparency in areas where there are few meteor detections and thus the estimate uncertainty is relatively high (i.e. the improvement in posterior predictive variance over the prior variance is less than 4 dB).

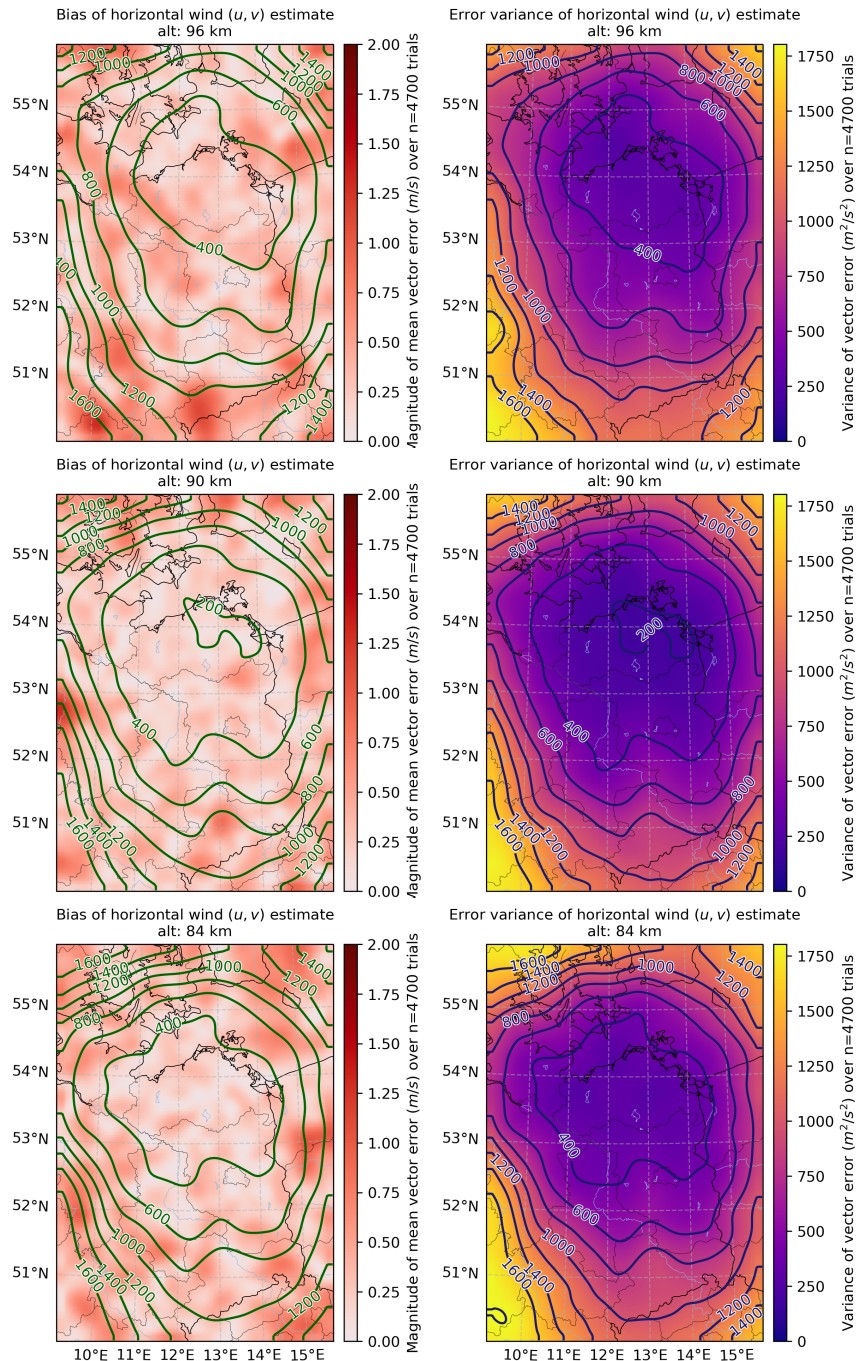

**Figure 3.** *Statistics of the horizontal wind estimator error relative to the simulated truth.* Each panel shows the bias (left) or error variance (right) as a function of latitude and longitude averaged over $n = 100$ trials at each of 47 measurement geometries taken throughout one day. Contours on the bias plots give the posterior predictive variance in units of $\mathrm{m^2\,s^{-2}}$, indicating more confidence in the central areas where the bias also tends to be a little lower. Contours on the error variance plots correspond to the sample error variance (matching the coloring).

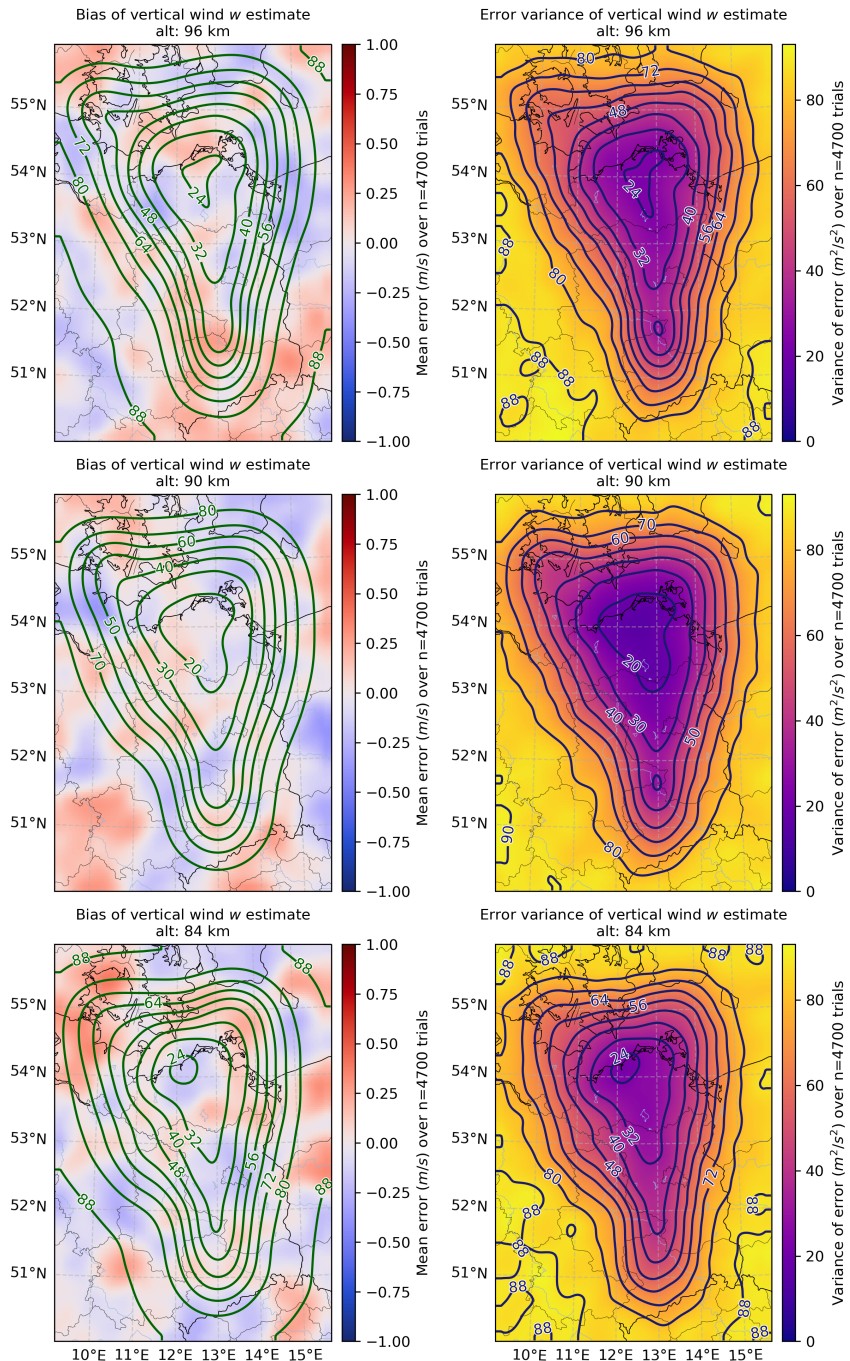

**Figure 4.** *Statistics of the vertical wind estimator error relative to the simulated truth.* Each panel shows the bias (left) or error variance (right) as a function of latitude and longitude averaged over $n = 100$ trials at each of 47 measurement geometries taken throughout one day. Contours on the bias plots give the posterior predictive variance in units of $\mathrm{m^2\,s^{-2}}$, indicating more confidence in the central areas where the bias also tends to be a little lower. Contours on the error variance plots correspond to the sample error variance (matching the coloring).

Similarly, the bias and variance results for the vertical wind are shown in Fig. 4. Again, we see low biases that are uniformly less that $1\,\mathrm{m\,s^{-1}}$ in magnitude, with the lowest biases in the regions of low predictive variance. However, this region is smaller than in the horizontal wind case. We are certain that this difference is mainly due to the configuration geometry that is needed to get accurate vertical winds, and the low-variance region provides a better observing geometry than the rest. Given the differences in magnitudes and the typically observed Bragg vectors, vertical wind estimates are relatively less constrained and more susceptible to horizontal wind contamination. Again, as in the case of the horizontal wind results, the uncertainty contours (left) roughly match the shape of the actual error variance (right).

## 5.3 Effects of scaling the covariance amplitudes

Until now we have presented results using estimator prior covariance amplitudes equal to the simulated values. In Figures 5 and 6, we show the biases and error variances while varying over different values of the estimator covariance amplitudes: (a) half, (b) equal, and (c) double the true value of the simulated winds. Specifically, we took the same 47 observation windows as before, simulated 100 random trials of measurements using covariance amplitudes of $\sigma_u^2 = 900\,\mathrm{m^2\,s^{-2}}$, $\sigma_v^2 = 900\,\mathrm{m^2\,s^{-2}}$, and $\sigma_w^2 = 90\,\mathrm{m^2\,s^{-2}}$, and estimated the winds with 9 different covariance amplitude combinations by scaling the horizontal and vertical values separately by $\frac{1}{2}$, 1, and 2. Note that the horizontal amplitudes for the zonal and meridional wind components were varied together such that $\sigma_u^2 = \sigma_v^2$. Finally, we computed the error between the estimated and simulated winds, calculated the mean and variance of the error over the random 100 trials (to give bias and error variance, respectively), and plotted the resulting distributions taken over time-space grid coordinates.

Figure 5 shows the GPR bias statistics for the zonal (top), meridional (middle) and vertical (bottom) wind components, with columns corresponding to halved (left), equal (center), and doubled (right) covariance amplitudes for the given wind component. The remaining vertical/horizontal covariance amplitude value is indicated with different colors. The salient features of this figure are: (a) the mean error has a tight distribution around zero, indicating little or no bias regardless of covariance amplitude scaling; and (b) the differences from scaling the covariance amplitudes are minor, with a slightly tighter bias distribution for the vertical wind component with a doubled vertical amplitude and halved horizontal amplitudes.

The posterior predictive uncertainties are plotted against the error variance in Fig. 6 for both the horizontal (left) and vertical (right) wind components. In the horizontal case, we show the results of the total horizontal wind speed, i.e., $\sqrt{u^2 + v^2}$. Lines give the mean of the error variance distribution, while the shaded region indicates the 90% confidence interval. For the horizontal/vertical wind plot, different line styles and labeling indicate the estimator values for the horizontal/vertical covariance amplitude while different colors indicate values for the vertical/horizontal covariance amplitude, respectively. The estimator covariance amplitudes match the simulated covariance amplitudes at the middle-orange values shown ($\sigma_u^2 = \sigma_v^2 = 900\,\mathrm{m^2\,s^{-2}}$ and $\sigma_w^2 = 90\,\mathrm{m^2\,s^{-2}}$), and those cases show good linear agreement between uncertainty and error variance. Halving and doubling the prior covariance amplitude of a given wind component similarly scales the posterior estimator uncertainty, resulting in either under- or over-estimating the uncertainty relative to the observed error variance.

Based on these Monte Carlo simulations, we recommend one of two approaches for applying GPR depending on the requirements of precision. First, if computational speed is a constraint and relatively large uncertainties are acceptable, then

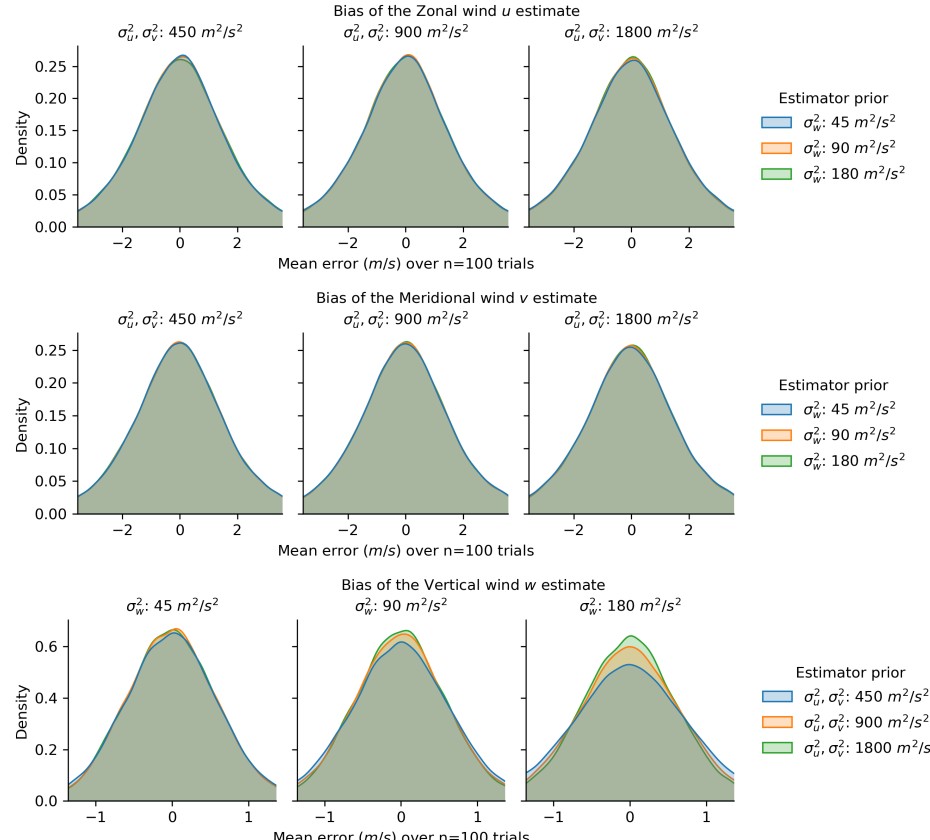

**Figure 5.** *Mean estimator error relative to the simulated truth when varying the covariance amplitudes.* Each panel shows distributions of the estimator error averaged over $n = 100$ random trials, where the distribution is taken over estimates at time-space grid coordinates where the estimated uncertainty shows meaningful improvement (defined as 1.5 dB). Relative to the simulated values, the estimator covariance amplitudes were scaled by $\frac{1}{2}$, 1, and 2 to test nine different combinations by varying values for both the horizontal ($\sigma_u^2 = \sigma_v^2 = [450, 900, 1800]$ $\mathrm{m^2\,s^{-2}}$) and vertical ($\sigma_w^2 = [45, 90, 180]\,\mathrm{m^2\,s^{-2}}$) wind components.

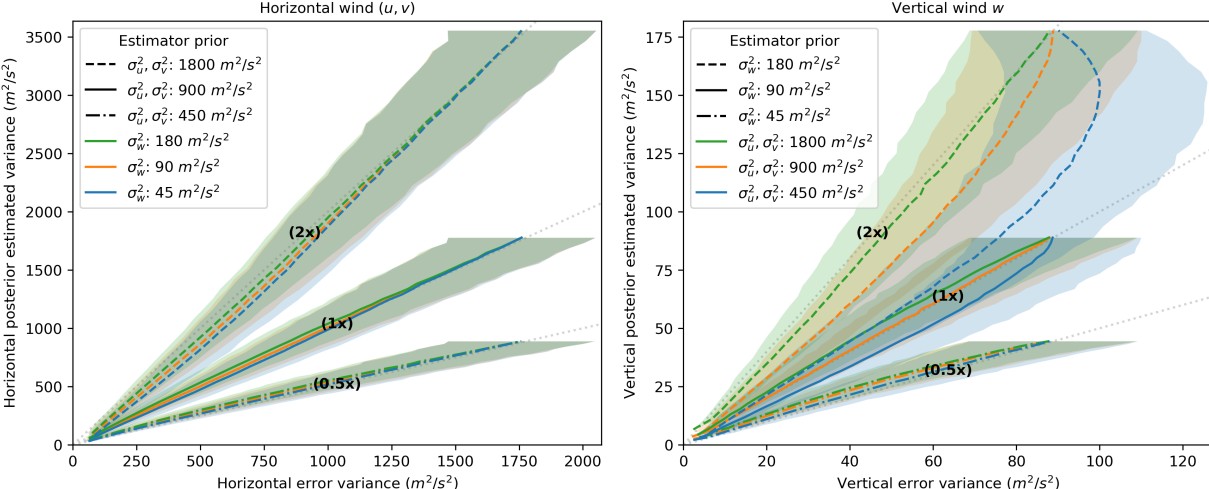

**Figure 6.** *Estimator posterior uncertainty versus error variance relative to the simulated truth when varying the covariance amplitudes.* Each panel plots the mean (lines) and 90% confidence interval (shading) of the distribution of the posterior predictive variance versus the error variance calculated over $n = 100$ random trials, where the distribution is taken over individual estimates at time-space grid coordinates. Relative to the simulated values, the estimator covariance amplitudes were scaled by $\frac{1}{2}$, 1, and 2 to test nine different combinations by varying values for both the horizontal ($\sigma_u^2 = \sigma_v^2 = [450, 900, 1800]\ \mathrm{m}^2\,\mathrm{s}^{-2}$) and vertical ($\sigma_w^2 = [45, 90, 180]\ \mathrm{m}^2\,\mathrm{s}^{-2}$) wind components.

using conservative overestimates of the wind variances to specify the covariance amplitudes will still yield unbiased wind estimates with uncertainties that can be treated as rough upper bounds on the error variances. Second, if more precision is needed and computational time is not a problem, then fitting on the incoming data to get more accurate estimates of the prior covariance amplitudes will yield unbiased wind estimates with more accurate uncertainties. This choice between specifying
the covariance hyperparameters and fitting for them is a critical decision for any user of the GPR method, as seen already in the block diagram of Fig. 1.

### 5.4    Qualitative role of the covariance length scales

We have not yet conducted a systematic study of the covariance length scales in the same manner as our examination of the covariance amplitude hyperparameters. This is both because the degrees-of-freedom in perturbing the values are greater, making the analysis more complex, but also because the length scales are easier to interpret without detailed analysis. Because
the model will enforce high correlation for coordinates that are "close" relative to the length scales, the covariance length scales set the effective resolution of the wind estimates. So intuitively, increasing the length scales will lose resolution and blur the estimates, while decreasing the length scales will gain resolution at the cost of increasing uncertainty (due to fewer measurements having a strong effect at a given estimation location). This intuition matches with the informal testing that we
have done in perturbing the length scales from the fitted values.

We have found that fitting the length scale hyperparameters generally does a good job of maximizing resolution while maintaining a usefully low posterior predictive variance. Those optimal values are determined by both the true covariance length scales of the wind field and the spatiotemporal density of the meteor measurements. For this simulated data, we know that the measurement density can support smaller length scales because the fitted values for the corresponding real data are roughly half for the $x$, $y$, and $t$ dimensions (see Sect. 6) compared to the values for the simulated winds. Nevertheless, fitting the estimation hyperparameters to the simulated data produced length scales close to the simulation values, showing that the fitting is responsive to the "true" wind covariance distances and does not just tune to the meteor measurement density.

As an alternative to fitting, one always has the option of setting the covariance length scales according to a desired estimation resolution. This is useful when one is content with sacrificing potentially better resolution for the sake of computational simplicity. In the case that the measurement density is not high enough to support analysis at those fixed length scales, that fact will be made clear by having few or no regions of low posterior predictive variance for the resulting winds. The estimates will likely not have the overall best uncertainty, but they will still be valid and thus useful.

## 6 Experimental Results

In this section we implement the proposed wind field estimator on a data set of 24-hour observations collected on November 5, 2018 during the SIMONe2018 campaign. After initial data quality control, almost 200,000 meteor detections were obtained in 24 hours. Using a conservative approach and performing further quality checks yielded 100,000 high quality detections. The filter criteria used in this second reduction required that detections were (a) within three standard deviations of the zero-order residuals, and (b) more than $30°$ above the horizon, to ensure that good interferometric angle of arrival (AOA) or angle of departure (AOD) estimates were obtained (e.g. Chau et al., 2019). Filtering by a minimum elevation angle also has the effect of ensuring that the errors in AOA/AOD, when projected into the vertical direction, have limited effect on the estimated altitude. Meteor location errors are not incorporated into the current GPR method, so their effect must be limited by ensuring that any potential coordinate deviations are much smaller than the covariance length scales used.

Subsequently, GPR results were obtained by first determining mean wind functions by fitting a 6-knot (altitude) by 6-knot (time) tensor product cubic spline over the entire 24 hours of data. The 12 spline parameters were calculated by solving the standard least squares problem completely independently of the GPR model. Then the covariance fitting procedure was applied on overlapping 90-minute windows spaced at 30 minute intervals to estimate the covariance amplitudes and length scales as they varied throughout the day. With the current procedure that computes the full covariance matrix, limiting to short time intervals like this is necessary for computational feasibility. The hyperparameters were found to be constant enough throughout the day that approximate overestimates would suffice and allow proceeding with a single set of hyperparameters. The resulting covariance hyperparameters are: $\sigma_u^2 = \sigma_v^2 = 900 \text{ m}^2\text{s}^{-2}$, $\sigma_w^2 = 90 \text{ m}^2\text{s}^{-2}$, $\delta_x = \delta_y = 26$ km, $\delta_z = 3$ km, and $\delta_t = 900$ s. Finally, the wind estimates were produced by selecting a fixed time, gathering data from the 90 minute window around that time (more than enough given the time length scale of 15 minutes), and computing the posterior predictive values at chosen spatial points.

To get a sense of the scales resolved with the GPR method, Figure 7 shows latitude-longitude slices of wind fields at three different altitudes (84, 90, and 96 km) and three different times (05, 08, and 11 UT). The presentation format is similar to Fig. 2, i.e., horizontal wind speeds are color-coded, and streamlines show the direction of flow. Areas of large velocity variance are shaded with 50% transparency to white. The wind fields show significant complexity, much more than can be well represented by the single mean vector per plot that would be reported by a monostatic meteor radar. On simple inspection, horizontal wind structures of $\sim$ 20-50 km are successfully resolved, which is commensurate with the horizontal length scale hyperparameter of 26 km.

In Figure 8, altitude-time slices at selected latitude-longitude points are shown for both zonal (left) and meridional (right) wind components. The large-scale tidal features are in good agreement with those obtained with the homogeneous method applied to the same data (see, Vierinen et al., 2019, Figure 6). The winds show significant variation between horizontal locations as expected.

Although we do not have a ground truth in this analysis to validate the horizontal scales we are resolving, we conduct an additional comparison to complement earlier identification of the large scale features (i.e., tides). In Figure 9, we compare GPR wind fields with those obtained with the homogeneous method (i.e., independent of latitude and longitude), and those obtained with a gradient method. Specifically, the homogeneous method uses a zero-order Taylor expansion, while the gradient method uses a first-order Taylor expansion. Both estimates have been obtained with altitude and temporal bins of 4 km and 4 hour respectively, in order to produce a good representation of large scale features. The specifics of the two methods can be found in Chau et al. (2017) and Chau et al. (2021), respectively.

The gradient wind fields are shown in the first row of Fig. 9 for three selected altitudes (84, 89 and 94 km). The arrows are color-coded with the horizontal wind speed (green tones), while the mean vertical wind from the gradient method is color-coded with red-yellow-blue tones. In the second row the GPR 3D wind fields are displayed in a similar manner to the gradient estimates in the first row. The third row shows the difference between the GPR wind fields and those from the gradient method. Note that the arrow colors and colorbar in the third row are different from the first two rows and show the difference of the horizontal winds. In all three rows the horizontal wind from the homogeneous method is shown with a thick black arrow in the center.

The salient features of Fig. 9 are:

- In general, there is good agreement in the horizontal wind components between the gradient and GPR methods. Note that the gradient estimates have been obtained with relatively large temporal and vertical averaging, in order to produce a good representation of large-scale features.

- By subtracting the mean wind obtained with the gradient method (i.e., large scale features) from the GPR estimates, in the third row, mesoscale structures are identified. Horizontal structures in the order to 20-50 km are clearly identified in all three altitude cuts.

Similar wind field comparisons for different times of the day can be found in supplemental material Movie S1.

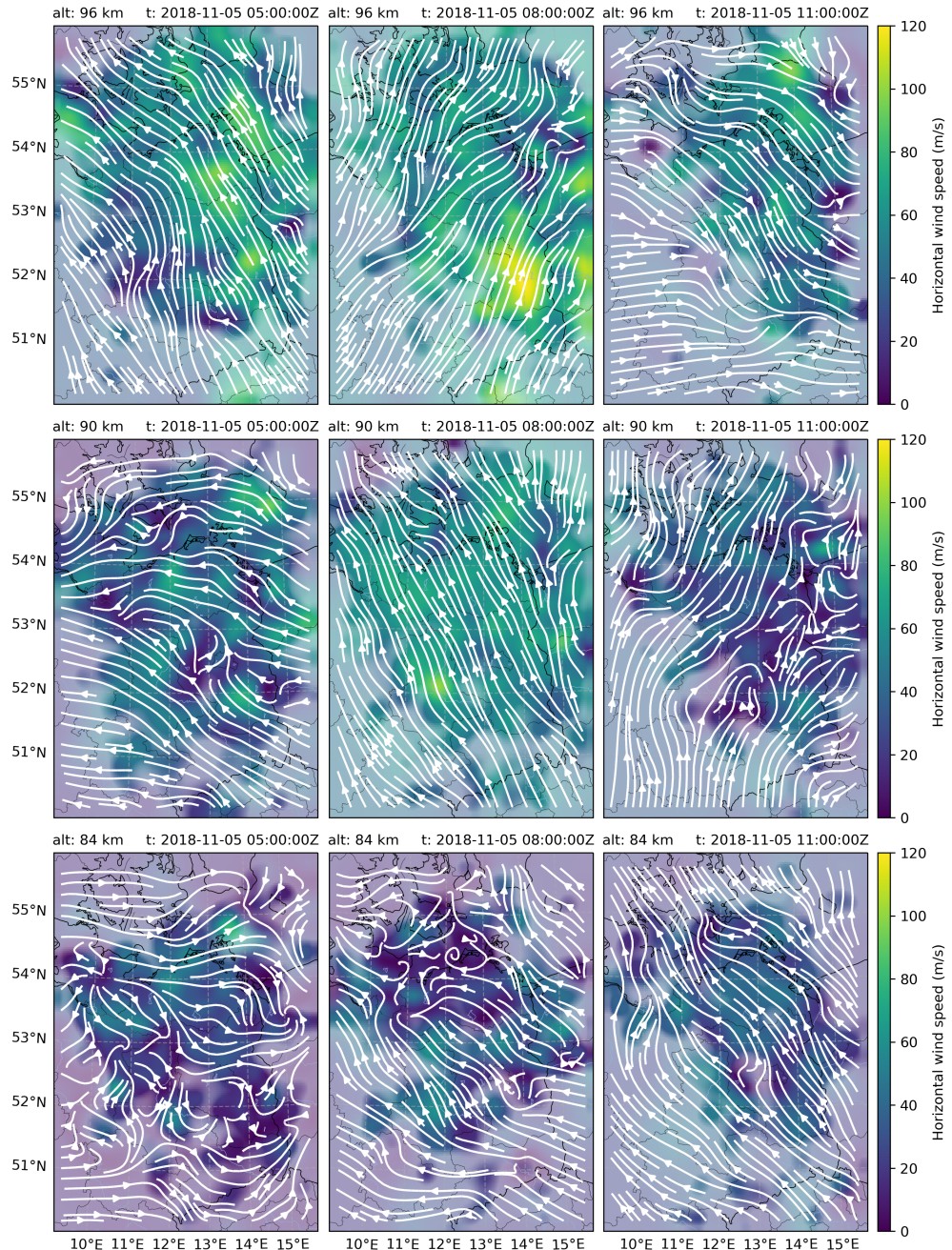

**Figure 7.** *Latitude-longitude slices of the winds estimated from SIMONe campaign data.* Each panel represents a separate altitude and time and shows the horizontal wind speed as a function of latitude and longitude overlaid by streamlines which show the wind flow. The wind speed is shown with 50% transparency in areas where the estimate uncertainty is large ($< 2$ dB improvement relative to prior uncertainty, i.e. where there are few meteor detections).

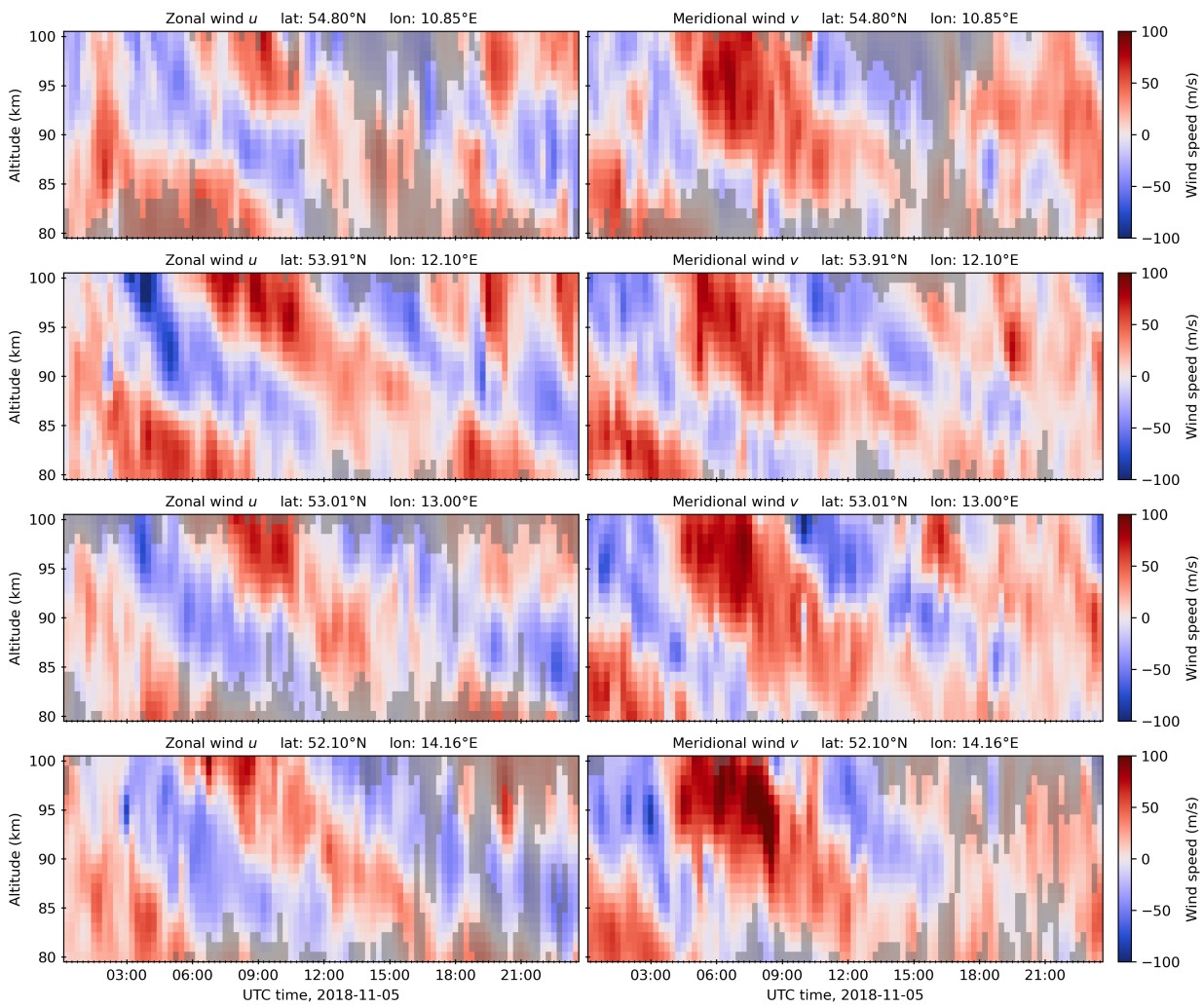

**Figure 8.** *Altitude-time slices of the winds estimated from SIMONe campaign data.* Zonal and meridional winds are shown at a selection of four latitude-longitude points. The wind speed is shown with gray shading in areas where the estimate uncertainty is large ($< 1$ dB improvement relative to prior uncertainty, i.e. where there are few meteor detections).

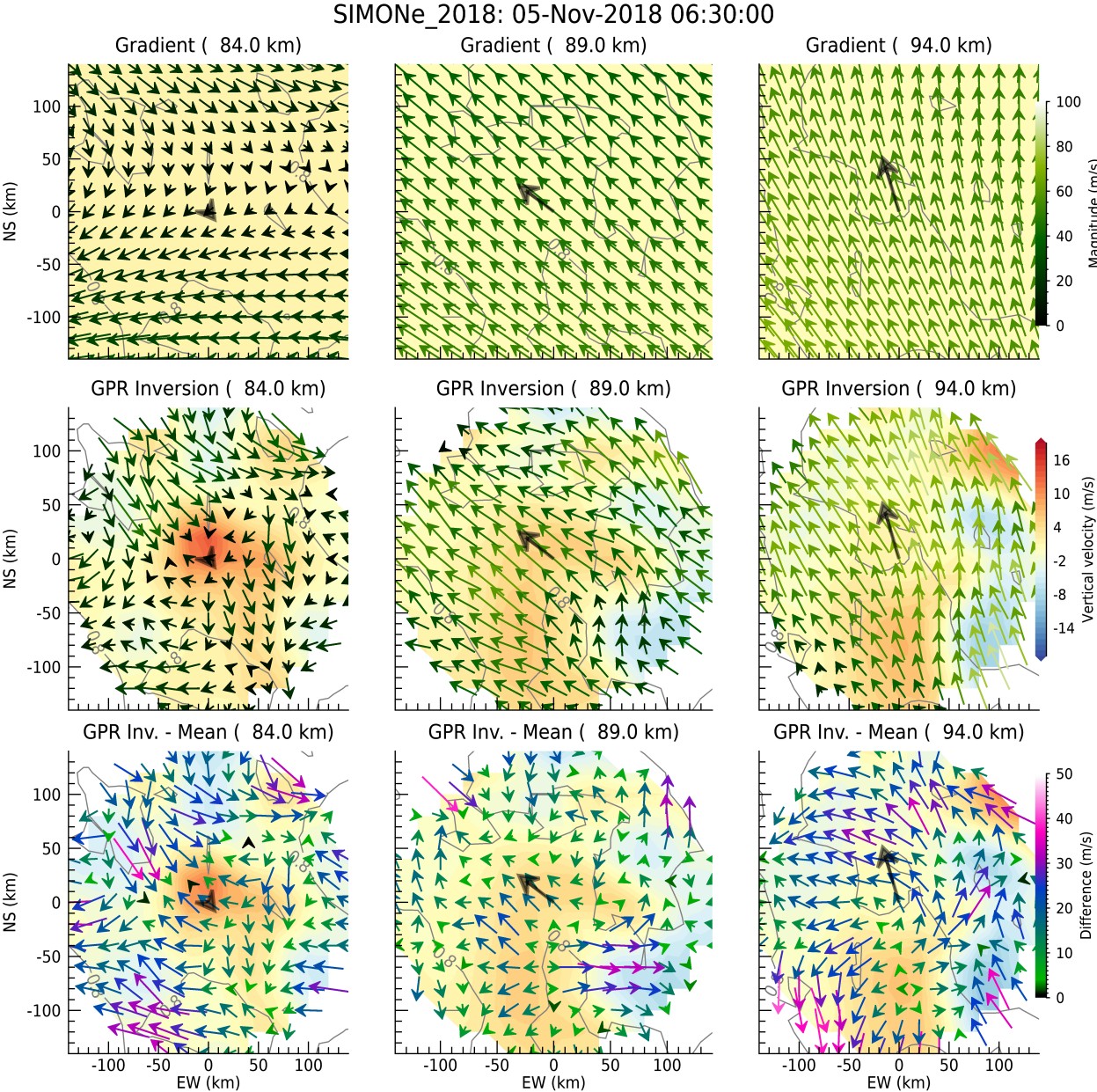

**Figure 9.** *Comparison of GPR to gradient and homogeneous methods.* The first row shows the horizontal wind field obtained with the gradient method using 4-hr and 4-km bins; the second row shows the horizontal wind field obtained with the GPR method using fitted covariance hyperparameters; and the third row shows the wind field difference between the values in the second row and the mean horizontal wind indicated in all panels with a black arrow. In all cases, a normalized statistical variance is indicated as gray contour lines, while the color contour represents the vertical component from gradient method (first row), GPR method (second row), and GPR minus mean from gradient method (third row). The row two color bar corresponds to the background vertical wind coloring while the other two color bars correspond to their respective arrow colors.

## 7 Discussion

We have introduced a robust method based on Gaussian process regression analysis to estimate MLT wind fields in four dimensions. The method has been evaluated using Monte Carlo simulations and implemented successfully on real data. The fast implementation using specified covariance hyperparameters (per-component amplitudes and per-dimension length scales) provides unbiased estimates with estimated uncertainties proportional to the prior velocity variances. In other words, if the prior variances are underestimated, the posterior variances are also underestimated. Using a more resource intensive training and fitting approach, covariance amplitudes can be estimated, resulting in posterior variances that are in good agreement with expectations from Monte Carlo simulations. The training approach requires more computation time than using fixed prior variances and we have not routinely applied it in analysis to date. However, for method testing purposes, we have implemented it on the real data shown in this work.

As expected, we have shown that mean values of GPR wind fields are in good agreement with the mean winds obtained with the homogeneous method. Similarly, to a first order approximation, GPR wind fields are also in good agreement with the wind fields obtained with the gradient method. Based on the simulation results, we expect the differences (i.e. the 20-50 km scales within their posterior variances) to be of geophysical nature.

Although the GPR method is robust, its region of validity and resolution depends highly on the geometrical configuration used, which influences the location and density of meteor observations and the observable projected wind component. For example, we found that the region of low variance vertical winds is smaller than the region of low variance horizontal winds. This result occurs even though the SIMONe2018 configuration has far superior properties in terms of links and diversity of Bragg angles compared to any other multistatic configuration used to date to study MLT winds (e.g., Chau et al., 2017; Stober et al., 2018; Spargo et al., 2019; Chau et al., 2021; Conte et al., 2021). Fortunately, the posterior predictive variances provided by the GPR method can be used in the future to optimize the meteor radar network geometry to achieve a given prediction goal, e.g. covering a specified region so that the estimate uncertainty for the winds reaches a particular value given typical meteor statistics.

Estimating the vertical wind component is still challenging due to two factors: the horizontal wind variability is larger than the vertical wind variability (leading to large contamination of the vertical wind when there are errors in the estimated Bragg vector or meteor location), and the majority of Bragg vectors have angles that are not close to zenith. The absence of zenith oriented Bragg vectors is intrinsic to all specular meteor radars, since any Bragg vectors with angles close to Zenith would require meteor trajectories parallel to the Earth's surface and are therefore very unlikely to be observed. In the particular case of the gradient method, Chau et al. (2017) have previously shown that the mean vertical velocity obtained with the homogeneous method, i.e., an area of $\sim$200 km radius, was contaminated by the mean horizontal divergence. Similar effects would be expected at smaller scales. Our experimental results do produce a vertical wind prior variance of about $90 \, \mathrm{m^2 \, s^{-2}}$, and some of the vertical wind estimates do show non-zero vertical velocities congruent with that variance. However, the posterior error bars are still large enough that a zero or nearly-zero vertical wind is a plausible explanation, especially considering the possible role of horizontal contamination. The important points relevant to the technique are that GPR is agnostic to the prior assumptions

one wants to employ for the vertical winds, and it also provides the necessary uncertainty information to allow for assessing the quality of the vertical wind estimates.

These results represent just the first step toward applying GPR analysis to estimate wind fields from meteor observations. We envision multiple directions of future work to expand and improve on the technique. There are many degrees of freedom in specifying mean and covariance functions to represent the wind components that can be explored. Known physical processes imply more structure in the joint wind component covariance than expressed in (21), so it would make sense to experiment with adding cross-covariance terms and allowing independent length scales for each component. The spatiotemporally varying sampling density imposed by the meteors argues for using covariance functions or hyperparameters that also vary in time and/or space. This can already be achieved in a crude form by performing fitting and estimation on overlapping subsets of the data, and we would like to explore that more as well as develop a more elegant approach. We've used the mean functions to essentially remove large-scale tidal effects, but it remains to be seen how to strike the optimal balance between complexity in the mean versus covariance functions or even the model complexity overall. At some point, adding complexity transforms the GPR method from data-based estimation into assimilative modeling, and we see value in prioritizing simplicity and clarity.

Incorporating the uncertainty in the meteor locations and Bragg vector components into the GPR analysis is another important avenue for improving the technique. We have so far removed any low-quality meteor detections from the analysis to limit the effect of this additional error, and the quality of the wind estimates would be improved by being able to incorporate this discarded data and make even better use of the high-quality detections. We anticipate that such a task would be challenging; it would likely entail leaving the closed-form solutions behind and numerically sampling from the distributions (e.g. Markov chain Monte Carlo methods).

Future work will also concentrate on further validation (including cross-validation within a single dataset), although it remains that currently no alternative MLT wind instrument is available for comparison with GPR estimates. Therefore, independent of the good comparisons with Monte Carlo simulations, we are planning to conduct special future observing campaigns under different atmospheric conditions and geometric configurations to intercompare our GPR method with other wind field methods such as those employing Tikhonov regularization (e.g., Stober et al., 2018; Chau et al., 2021). Similarly, we plan to compare these techniques using synthetic data from regional weather models with high resolution covering the MLT alitudes, such as the ICON-UA model (e.g., Borchert et al., 2019). This analysis concept would be similar to the one implemented in this work, but with more realistic atmospheric dynamics for the simulated winds.

Finally, we plan to apply the GPR method to selected additional datasets that use a multi-static configuration in order to further investigate the properties of the resolved 20-50 km horizontal wind structures. These investigations will cover both individual case studies and statistical studies: for the former, we expect to analyze special geophysical conditions and/or measurements that are complemented by other ground- or satellite-based instruments (e.g., Davis et al., 2018; Vargas et al., 2020); for the latter, we expect to compare the Reynolds stress tensor statistics of GPR-estimated wind fields to those obtained from second-order statistics of projected wind velocities (Vierinen et al., 2019).

## 8 Conclusions

We have introduced an alternative observation method based on Gaussian process regression analysis to resolve MLT wind fields in 4D from multistatic radar observations. Based on Monte Carlo simulations of known wind field distributions, our proposed method provides unbiased mean velocity estimates and posterior velocity variances that are proportional to prior velocity variances. By using an adaptive fitting procedure based on input data, unbiased posterior variances can be achieved. This adaptive approach is currently not practical for real-time applications, but is ideal for case studies.

The horizontal regions of good GPR method performance in MLT wind determination are dependent on the meteor scatter geometric configuration. On one hand, optimal configurations should ultimately increase the number of detections. However, on the other hand, these same configurations need to provide sufficient Bragg vector diversity. For the particular SIMONe2018 experiment scattering geometry, these factors meant that vertical velocity estimates with relatively small variances were obtained over a much smaller horizontal area than horizontal wind estimates.

Overall, the GPR method has attractive benefits for MLT regional and weather studies: 1) it enables flexible analysis by allowing grid-free wind estimates; 2) it provides statistical uncertainties for the estimated winds that reflect measurement uncertainty and meteor observation geometry; and 3) it adapts to the horizontal, vertical, and temporal scales of the data, accounting for measurement density, and thus is able to resolve winds at relatively small scales.

*Data availability.* Meteor observations from the SIMONe 2018 campaign on November 5, 2018 and wind estimates produced by the GPR
method can be found at https://zenodo.org/record/5550854 (Volz et al., 2021). Additional information and hyperparameters used for the GPR wind estimates can also be found there.

*Author contributions.* RV conceived of and refined the Gaussian process regression wind estimation approach through discussions with JLC, PJE, JPV, and JMU. RV implemented the technique and performed the formal analysis. MC performed the meteor estimation, curated data, and wrote software that was used in the analysis. JLC, PJE, and RV wrote most of the paper.

*Competing interests.* The authors declare that they have no conflict of interest.

*Acknowledgements.* This material is based upon work supported by the National Science Foundation under Grant Nos. 1626041 and 1933005. This work was partially supported by the Deutsche Forschungsgemeinschaft (DFG, German Research Foundation) under SPP 1788 (CoSIP)-CH1482/3-1. The authors gratefully acknowledge the support of an international team from the International Space Science Institute (ISSI-Bern) and discussions within the ISSI Team 410. The authors would like to thank everyone who contributed to the SIMONe

2018 campaign: Nico Pfeffer and Jörg Trautner for designing and implementing the hardware and operational software, and Fede Conte for supporting operations.

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
