# Peer review of "Four-dimensional mesospheric and lower thermospheric wind fields using Gaussian process regression on multistatic specular meteor radar observations"

_Atmospheric Measurement Techniques, 2021_

## Community Comment (CC1)

[Figure]

Figure 1: Top: Illustration of a trail and its echo location under top:wholly horizontal wind condition. bottom: for wholly vertical wind.

I strongly disagree with the basis of this paper. The basis as stated at the begining of section 3 is that the measured radial velocity represents the component of actual atmospheric motion in the radial direction. In fact for specular reflection from a line reflector, what is seen is the radial motion of the reflection point. The reflection point is the intersection between the line reflector and the line of sight. This is only the atmospheric motion if the line reflector (meteor trail) is horizontal. Otherwise the difference depends on the trail orientation (rotation) with respect to the line of sight

For example, imagine that the wind has no vertical component Then any trail seen above the horizon cannot have a vertical component of motion, but the radial velocity appears to have one. That is, vertical and horizontal are inextricably linked.

The figure shows this graphically

---

## Author Response (AR1)

**Author's Response - AMT-2021-40**

We wish to thank all of the reviewers for their thoughtful and detailed comments. They have been very useful in helping to clarify the paper.

**Response to RC1**

*1. Throughout the paper, the GPR method is described as being non-parametric. However, the prior model for the winds is necessarily parametrized by the means and variances of the wind components (theta). In what sense is this method non-parametric?*

We describe GPR as non-parametric in the same sense as it is used in the referenced Rasmussen and Williams (2006) book: an estimation method which does not compress the training data into a finite-dimensional parameter vector, in contrast to parametric methods like linear regression. The mean and covariance function parameters are usually called hyperparameters to emphasize that they are parameters of a non-parametric model. We also find this terminology confusing, so we have clarified this point in the revised manuscript.

**Changes in manuscript:** We have changed previous references to the model's parameters to refer instead to "hyperparameters", and we have included a sentence in the introduction explaining what is meant by "non-parametric".

*2. The paper distinguishes the GPR method from Tikhonov regularization which is viewed as a competing method. However, 2nd order Tikhonov regularization can be interpreted as the adoption of a Gaussian prior for the state vector. How is this fundamentally different from the method presented here?*

The methods are indeed related, and there is a good discussion of this topic throughout Rasmussen and Williams (2006), particularly in Chapter 6. The most direct difference is that Tikhonov regularization would best relate to GPR with a squared exponential covariance, whereas we have employed a Matern-5/2 covariance. That detail strikes at the heart of the difference between the two methods: it can often be more natural to express prior knowledge in terms of a GPR covariance than as a likelihood penalization. Practically, the difference also comes down to how it is more natural to perform non-gridded estimation and analyze uncertainty with GPR compared to regularization approaches. In many respects, the approaches are two sides of the same coin, which is why we see value in future inter-comparison that can help refine both approaches.

**Changes in manuscript:** We have added a paragraph in the introduction to expand on the discussion of GPR in general, including a note that it is highly-related to interpolation techniques that employ regularization. We mention why one might choose GPR over these other techniques. We also refer the reader to Rasmussen and Williams (2006) for an extensive discussion of these topics.

*3. The paper describes the assumption of a Gaussian random process for the winds as "convenient" because of the tractable computations that result. Could not the authors provide a more satisfying rationale by considering whether the central limit theorem applies to the MLT wind components?*

This question prompted us to think more about this assumption, so we thank you for that. Assuming normality imposes the minimal prior information about the wind processes within a second-order statistical framework since the Gaussian distribution has maximum entropy for a known mean and covariance.

**Changes in manuscript:** We have added an explicit justification for the Gaussianity assumption to Sect. 3.2.

*4. The authors of the paper note the absence of other kinds of MLT wind measurements that could be used to validate the wind estimates produced by the GPR method. Have the authors considered the use of generalized cross validation?*

Generalized cross-validation is indeed one technique that could be used to further assess the validity of the wind estimates using just the data we already have. There is a good discussion of cross-validation in the context of GPR in Rasmussen and Williams (2006), section 5.4. We think this is a good area for future work (and will additionally note it in the revised manuscript), and the result would be a better understanding of the covariance hyperparameter values and the merit of different forms for the prior wind distributions. We leave it to future work because it addresses the topic of improving the model specification, and that area is large enough to merit its own paper(s) beyond the introduction of the technique that we provide here. Although such work would speak to validation, it still remains that comparison to a completely independent technique would go even further to give confidence in the GPR method.

**Changes in manuscript:** We added a parenthetical mention to performing cross-validation as a subset of future validation work.

*5. The authors note that the region of validity and the spatial resolution of their method depends on the geometrical configuration of the multistatic meteor array. Have they considered developing a geometrical dilution of precision (GDOP) estimator?*

Perhaps it is clearer to say that the measurement geometry controls the wind estimate uncertainty, which we can calculate through the posterior covariance, and naturally there are (location, wind direction) pairs that have higher uncertainty. As far as we understand it, we can quantify the GDOP as a function of location in this case by taking the root mean square of the wind component uncertainties.

**Changes in manuscript:** None

*6. Finally, interpreting figures 2--4 and 7 is very difficult in view of the fact that color gradations are being used to represent multiple quantities simultaneously (i.e. horizontal winds, vertical winds, and data quality). The authors should attempt to clarify these figures.*

We very much would like these figures to be both expressive and interpretable, and we recognize that this is a difficult task given the amount of information that they attempt to include. We strived to use color scales for the different elements that are distinct enough to be identifiably separate and thought we had achieved a good balance. Considering this and other reviewer and community comments, we have simplified the figures somewhat in the revised manuscript. Notably, we have removed the vertical wind coloring from the horizontal wind streamline maps. We decided that the focus of these figures should be the horizontal winds, and including the additional color axis invites confusion and diverts focus. Additionally, we have simplified the bias panels in Figures 3 and 4 by removing the green shading and changing the contour lines to depict the predictive variance values instead of dB improvement. This makes the comparison to the error variance more direct, while still giving the viewer the necessary information to focus on the more pertinent regions of the bias values.

**Changes in manuscript:** We have updated Figures 2, 3, 4, and 7 as described above.

**Response to RC2**

*1. Based on the description of the GPR algorithm to estimate winds at any particular point in space and time, it is necessary to provide the mean and covariance matrix of the a priori wind distribution. While there is a nice description about how the covariance matrix can be modeled as a Matern-5/2 covariance function, there is no much discussion about how to determine the prior mean wind. In the manuscript, it is mentioned that it is obtained from applying a tensor product cubic spline to the dataset. However, it is not clear, at least to me, if the authors are calculating this mean from wind estimates that were obtained applying a different method, for instance with the zero-order method. Please clarify this point. In addition, the role of the prior covariance is analyzed in detailed, but the role of the mean is not. I can imagine that the results have a strong dependence on the a priori mean that is provided. The errors will probably increase depending on the accuracy of the mean. I would also suggest to discuss about the role of the a priori mean wind field in the manuscript.*

We concentrated on specifying and analyzing the prior covariance function because it is by far the more important component in the GPR specification, but we agree that more can and should be said about the prior mean function. The biggest effect of specifying a good mean function is that it improves the covariance function specification, allowing the amplitude and length scale hyperparameters to be smaller in general. This leads to smaller error bars on the wind estimates, but the wind estimates themselves (perhaps surprisingly?) don't change too much.

**Changes in manuscript:** We have added additional discussion in Sect. 3.2 of how we specify the mean and how that choice affects the final estimates. We have also included a statement in Sect. 6 about how we solved for the spline parameters that form the mean function in our wind estimates.

*2. The position of the detected meteors are also the result of a fitting procedure, thus there are uncertainties associated to the space and time location of a meteor. How these uncertainties are taking into account in the GPR algorithm? Do they play any role in the accuracy of the estimated winds? I understand that a filter criteria is applied to the data to consider only high quality detections, what is the criteria that is used? Do the high quality detections have negligible uncertainties for the meteor locations? I would also suggest to discuss this question in the manuscript.*

Our current GPR method does not incorporate uncertainty in the meteor coordinates (space and time). We agree that it would be great to include this, but the GPR framework does not naturally incorporate this and adding it would be a significant project for future work. We expect this would entail leaving the closed-form solutions behind and numerically sampling from the distributions (e.g. MCMC). But for the analysis in this paper, we do try to limit the effect of coordinate uncertainty by throwing out low-quality detections. Overall the uncertainties for the high quality detections are small enough relative to the covariance length scales that the added estimation error is negligible.

**Changes in manuscript:** We have clarified the process of throwing out low-quality detections and its implications in Sect. 6. We have also added a paragraph in Sect. 7 to discuss incorporating this uncertainty in future work.

*3. The detection of meteors in time can be modeled as a Poisson process, in the sense that given a detection the probability of detecting the next meteor increases as function of time. Given this, the time location of the Doppler samples is also a Poisson process. However, for the GPR approach, we are assuming that the Doppler samples can be modeled as a Gaussian process in space and also in time. What is the impact of this difference in the estimation of the MLT winds?*

One must be careful to distinguish between the probability distribution of meteors occurrence/detection and the distributions used to model the winds. For the wind process, when and where the meteors occur is irrelevant; all we care about is that the meteors produce a set of measurements, and how those measurements are distributed does not factor into our assumptions about the wind processes. A practical effect of the Poisson distribution for meteor detections, however, does mean that our coordinate sampling of the winds is more grouped than it would be if the meteors were uniformly random. That just means that we'll get lower uncertainties for the wind estimates in those regions due to the abundance of samples.

**Changes in manuscript:** To help clarify this point in addition to RC1.3, we have added more discussion of the wind process distribution assumption in Sect. 3.2.

*4. While the role of the covariance amplitudes for the a-priori wind distribution are discussed in detailed, there is no much discussion about the role of the space and time scales. In principle, these parameters are also obtained from minimizing the negative log-likelihood, however, I can*

*imagine that their values will strongly depend on the distribution of data considered. For instance given some particular data set where data samples are more concentrated around 90 km but more sparse around 80 or 100 km altitude, it is reasonable to expect that the space scale values will also vary as function of height, they will be probably shorter round 90 km but longer at higher or lower altitudes. Is this something that is considered in the algorithm? How the wind estimations will be affected by the precision of the space scales used in the algorithm?*

This comment matches our experience with the length scales: the fitted values are strongly driven by the density of meteor sampling within a particular dataset, so we might naturally want to use smaller values around 90 km and during the morning detection peak and larger values at low/high altitudes and during the evening detection valley. This is not currently considered in our GPR technique, since we are using constant values for the length scales that don't vary with location. We think that allowing the length scales to vary with location (e.g. altitude) would likely lead to better wind estimates, and we think that this would be a fruitful area for future work (and have already suggested this in the manuscript, if not quite as clearly).

**Changes in manuscript:** We have added Sect 5.4 to discuss the qualitative role of the covariance length scales.

*5. In section 5.2, in the case of the horizontal wind, it is not clear whether the "mean bias error" or the "mean absolute error" were calculated. The labels in Figure 3 indicates "u+v", does this mean that the values of the zonal and meridional winds were added before calculating the error? Please, clarify this issue, the authors may consider to include the actual formulas used to estimate the errors. Also, in the same figure, the titles of the plots on the right side indicate "variance of horizontal wind", however, I think the authors are referring to the variance of the horizontal wind error. Please, fix the labels of the figure to clarify their meaning. Similar comment with respect to Figure 4, have the authors plotted the variance of the vertical wind or the variance of the error?*

We intend to say "magnitude of the mean error of the horizontal wind vector" (averaged difference in the horizontal wind vector, including both u and v components) for the bias plot, and we intend to say "variance of the horizontal wind *error*" for the variance plot. Same for the vertical winds.

**Changes in manuscript:** We have updated Fig. 3 color labels to refer to "magnitude of mean vector error" and "variance of vector error" and specified *error* variance in the titles and caption. We have similarly updated Fig. 4, except with reference to "mean error" and "variance of error". We have added a description of how we calculate the bias and error variance to Sect. 5.2.

*6. In section 6, in the implementation of the algorithm, it is mentioned that the data is divided in time intervals of 90 minutes with 30 minutes overlapping, would not this have an impact on the smoothness of the winds that are estimated? Would not it have sense to apply a similar criteria in space (altitude, latitude, and longitude) based on the actual distribution of meteors? The*

*accuracy of the wind estimates in the areas further from the center may improve if covariance parameters are computed particularly for these regions. This comment is related to the role of the space and time scales presented above.*

The overlapping estimation procedure is not a necessary component of GPR, but it is helpful for reducing the computational burden as long as care is taken. And by that we mean that we have only made estimates at times when at least 45 minutes of data both before and after are included, for a total time window of 90 minutes (with centered estimate). This window is wide enough, given the 15 minute time length scale for the covariance, to ensure that the estimates produced are only negligibly different from the result of if a wider time window (or the whole dataset) was used. We have verified this by comparing the 90-minute-window estimates to ones done with a 180 minute window. Thus, the smoothness of the wind estimates is not affected.

It is an astute observation that the overlapping estimation procedure could be used to apply different hyperparameters that are more tuned to different segments of the data. This is indeed the most straightforward way to apply that type of analysis for future work, even if it is not terribly elegant. This is also how we know that the density of meteor detections affects the hyperparameters: we have observed that the fitted length scales in particular change somewhat throughout the day (when fitted to these overlapping windows of data), seemingly in correlation to the density of meteor detections. Fortunately the estimates are not changed greatly by imposing constant conservative values throughout the day; it just means that we're not achieving quite the best resolution at times when the meteor density would support it, effectively smoothing over potentially-detectable features.

**Changes in manuscript:** We have added a brief mention in Sect. 7 of using overlapping data segments to explore varying the hyperparameters as a function of space and time.

*7. In Figure 9, Gradiente winds and GPR estimates are compared. In particular, it is mentioned that the GPR estimates show some mesoscale structure. Are the authors implying that the GPR method was capable of estimating these mesoscale structure while the other method could not do it? I would recommend to clarify this comparison. The gradient winds were computed considering wider time intervals, and probably that has an impact in the smoothness of the estimated winds, thus the comparison would not be fair.*

Perhaps it is not totally fair to make this comparison and the claim that GPR shows mesoscale structure where the gradient method does not. It is definitely possible to perform an analysis with the gradient method (or other existing methods) that focuses on time and length scales similar to the GPR method, and thereby likely identify the same mesoscale structures. Such information is in the data, and we don't mean to claim that GPR performs some magic that unlocks it that is inaccessible to other methods. The benefit of GPR is not necessarily that it allows one to see these mesoscale structures, but that it provides a suitable framework and procedure for identifying those scales within the data and making them clear without manual data analysis.

**Changes in manuscript:** We have removed the note that GPR identifies smaller horizontal spatial scales than the gradient method. We have also changed the final point of the conclusion to highlight the adaptability of the GPR method for analyzing different spatial scales and selecting appropriate scales based on the data, which is where it actually provides a benefit over the gradient method.

*8. In Figure 2 and Figure 7, the vertical winds are depicted as the color of the horizontal wind lines. In fact in the labels, it is mentioned "vertical wind speed", however, "speed" by definition is a positive value, and the colors go from positive to negative, so I think it is more appropriate to change the label to "vertical wind". In addition, are the values of the estimated vertical winds within the expectation? How do they compare with their corresponding variances? Are the error bounds small enough to have good vertical wind estimates? It would be important to add a discussion in the manuscript about the accuracy of the vertical winds estimated with the GPR method given that previous methods have just assumed that the vertical winds are zero.*

In response to this and other discussion of the vertical winds and the figures, we have decided to remove the vertical wind component from the Figures 2 and 7 to improve clarity. Nevertheless, we have added more discussion of the vertical winds to the manuscript to address the questions raised in this and other reviews. The basic conclusion is that the technique is agnostic to the prior assumptions the user wants to employ for the vertical winds, and it also provides the necessary uncertainty information on the wind estimates that will allow the user to assess the quality of the vertical wind estimates. Through the typical meteor observation geometries, there is much less information about winds in the vertical direction than the horizontal directions. The fitting procedure on the SIMONe dataset produced a prior variance for the vertical wind component of about 90 m^2/s^2 using a set mean of zero. This could be from actual instantaneous non-zero vertical wind values, but it could also be elevated due to errors in the Bragg vector direction and/or meteor location causing contamination from the horizontal winds. The values produced in the estimates conform to this prior distribution and the information added through the measurements, but the posterior error bars are still large enough that a zero or nearly-zero vertical wind is a plausible explanation, especially considering the possible role of horizontal contamination. Great care is still needed in this and any future analysis of vertical winds, but we think GPR will provide a useful new tool in performing that analysis.

**Changes in manuscript:** We have removed visualization of the vertical wind component from Fig. 2 and 7. We have added more discussion of the vertical wind components in Sect. 7 to highlight the challenges in estimating the vertical winds, emphasize that the GPR technique provides useful tools in performing that analysis, and state that a nearly-zero vertical wind would still be consistent with the estimates that we have presented.

*9. Finally, in the conclusion section, the authors indicate that the GPR method can resolved winds at the finest spatial and temporal scales allowable by the instrument. However, what are these finest scales allowable by the instrument? The geometry and the spate-time distribution of meteors will definitely set limits to the features that can be resolved both in space and time with*

*this method, however, aren't other methods within their assumptions also capable of resolving fine structures? In fact, in section 6, it is mentioned that the authors do not have a ground truth to validate the horizontal scales that are resolved by the GPR method. Given this it cannot be claimed that the GPR method resolves the finest scales allowable. I would recommend to change this conclusion.*

The discussion of comment RC2.7 is also relevant here. It is evident that we need to clarify the point we are trying to make with this discussion and conclusion.

**Changes in manuscript:** We have changed the conclusion to highlight the adaptability of the GPR technique and how that is helpful for resolving finer spatial scales.